# The rough sound of salience enhances aversion through neural synchronisation

Luc H. Arnal [1], Andreas Kleinschmidt[2], Laurent Spinelli[2], Anne-Lise Giraud[1] & Pierre Mégevand [1,2]

Being able to produce sounds that capture attention and elicit rapid reactions is the prime goal of communication. One strategy, exploited by alarm signals, consists in emitting fast but perceptible amplitude modulations in the roughness range (30–150 Hz). Here, we investigate the perceptual and neural mechanisms underlying aversion to such temporally salient sounds. By measuring subjective aversion to repetitive acoustic transients, we identify a nonlinear pattern of aversion restricted to the roughness range. Using human intracranial recordings, we show that rough sounds do not merely affect local auditory processes but instead synchronise large-scale, supramodal, salience-related networks in a steady-state, sustained manner. Rough sounds synchronise activity throughout superior temporal regions, subcortical and cortical limbic areas, and the frontal cortex, a network classically involved in aversion processing. This pattern correlates with subjective aversion in all these regions, consistent with the hypothesis that roughness enhances auditory aversion through spreading of neural synchronisation.

---

[1] Department of Fundamental Neuroscience, University of Geneva – Biotech Campus, Geneva 7 CH-1202, Switzerland. [2] Department of Clinical Neuroscience, University of Geneva – HUG, Geneva 14 CH-1211, Switzerland. Correspondence and requests for materials should be addressed to L.H.A. (email: luc.arnal@unige.ch)

A first and foremost purpose of communication is to catch the attention of conspecifics, a process that can be optimised by adapting signal salience to maximize the receiver's sensory-motor responses. To amplify sensory salience and ensure efficient reactions on the receiver's end, a generic strategy is to increase signal intensity, e.g. by screaming or shouting. However, signal magnitude is not the only parameter that changes when we increase vocal sound levels. Another important emerging feature is *roughness*, an acoustic texture that arises from fast repetitive acoustic transients. Although the delimitation of the roughness range—whether psychoacoustic or perceptual—may slightly vary depending on experimental settings[1–3], empirical observations consistently suggest that sensory systems and perception are exceedingly well tuned to recurring temporal features in the 30–150 Hz range[4–8].

Human sensory systems are not passive sensors or filters but instead display nonlinear properties that constrain the way we perceive and process incoming inputs[9–11]. As a result, sensation is not a linear function of physical features, but depends on neural coding transitions that determine perceptual categories or attributes[12,13]. In the auditory domain, the limited temporal resolution entails a subjective transition from rough percepts to continuous pitch sensations[2,14]. Despite its central position in the audible spectrum and recent efforts to uncover the underlying neuronal coding schemes[15,16], the neural bases of the transition in this frequency range and its consequences on subjective perception remain unclear[1].

Fast repetitive modulations produce "temporally salient" flickering percepts (e.g. strobe lights, vibrators, and alarm sounds[6]), which efficiently capture attention, generally induce rough and unpleasant sensations, and elicit avoidance[5]. Despite the high ecological relevance of such flickering stimuli, there is to our knowledge no existing operational definition of *temporal salience* and only limited experimental work accounting for the intriguing aversive sensation such auditory textures produce and the reactions they trigger. Here, we introduce and explore the notion of temporal salience and investigate its behavioural and neural underpinnings. Of note, although salience may not systematically result in aversive percept, we argue that in this specific context, temporal salience—owing to the imperative effect of exogenously saturating perceptual systems in time—constitutes a valid proxy of aversion. Therefore, we hypothesise that providing fast, but still discretisable and perceptible, temporally salient acoustic cues should enhance neural processing and ensuing aversive sensation. Such a strategy, however, is arguably constrained by the capacity of the auditory system to discretize—i.e. to faithfully encode and ultimately perceive—these temporal cues.

In this study, we assess the relevance of such temporal salience in human neurophysiology and its impact on sensation. In a series of psychoacoustic experiments featuring perceptual tasks, we determine that aversion to repetitive sound transients varies in a nonlinear manner with sound frequency and is maximal below the perceptual discretization limit, in the lower roughness range (40–80 Hz). Building upon this psychophysical characterisation of temporal salience, we then exploit intra-cortical recordings in patients with epilepsy to demonstrate that the aversion to rough sounds results from the sustained synchronisation of auditory networks, but also salience-related networks.

## Results

**Subjective characterisation of temporal salience**. To determine the subjective transition between discretisable (rough) and continuous (pitch) percepts, 16 participants listened to a series of 1-s click trains of varying rates (ranging from 50 to 250 Hz) and reported whether they perceived these sounds as discrete or continuous. The pattern of responses (Fig. 1a) consistently followed a sigmoid curve (goodness of fit: Pearson's $r^2 = 0.99$; $p = 10^{-12}$), evidencing that the subjective perceptual transition from rough to continuous—the *temporal discretisation limit*—occurs at frequencies exceeding $130 \pm 6.5$ Hz (mean ± SEM).

We then hypothesised that repeating sounds at frequencies close below the temporal discretisation limit should maximise the rate of discrete sensory responses across time, and in turn induce a more aversive percept. In other words, temporal salience—here reflected by subjective aversion—should depend on the auditory system's ability to discretize temporal fluctuations. We thus asked the same participants to rate the subjective aversiveness of 1-s click trains at varying rates (ranging from 10 to 250 Hz) from 1 (tolerable) to 5 (unbearable). Figure 1b reveals that aversion varied in a nonlinear, non-monotonic manner with increasing rates (Pearson $r^2 = 0.05$; $p = 0.2$). Qualitatively, this profile suggests that the 130 Hz transition evidenced in the previous study determines a bimodal effect on perceptual aversion across frequency rates. While subjective responses linearly increased as a function of stimulus frequency in the pitch (>130 Hz) range (goodness of fit: $r^2_{>130Hz} = 0.97$; $p = 0.01$), the response profile was nonlinear below this value (goodness of fit with linear extrapolation—see red line in Fig. 1b: $r^2_{<130Hz} = 0.007$; $p = 0.4$), increasing up to 40 Hz and then decreasing progressively up to the temporal discretisation limit (130 Hz). Click trains were perceived as particularly aversive in the 40–80 Hz range, decreasing below and above this range, up to the discretisation limit.

We noted significant variability across participants' ratings (grey circles in Fig. 1b). To test whether the nonlinear pattern of aversion might depend on sound intensity, we replicated the experiment repeating the exact same sounds presented earlier at lower intensity (~50 dB SPL). Although lower sound intensity reduced overall aversion ($F_{(1,20)} = 193.8$, $p < 10^{-7}$, Fig. 1b upper right inset), the pattern of ratings strikingly replicated the results obtained in the previous experiment (Pearson correlation: $r^2 = 0.86$, $p < 10^{-5}$, Fig. 1b lower right inset).

The fact that temporal salience (here reflected by subjective aversion) varies non-monotonically as a function of stimulus rate and is maximal in the roughness range suggests that nonlinear neuronal phenomena occur in this frequency range. To better understand the neural mechanisms underlying this effect, we measured the neural responses to click trains of various frequencies (2 s duration ranging from 10 to 250 Hz) using intracranial EEG (iEEG) in 11 patients with intractable epilepsy. We hypothesised that the initial auditory-evoked response should reflect the energy (i.e. the rate) of the stimulus[17], but that temporal salience should depend on whether the click trains induce sustained, steady-state entrainment of neural responses over time.

**Neurophysiological correlates of temporal salience**. To investigate basic auditory responses to these sounds, we measured the high-gamma (HG, 70–200 Hz) responses (as a proxy of neuronal firing[18]) in one individual iEEG electrode, which lays in early auditory cortex (anterior transverse temporal gyrus of Heschl, Fig. 2a). Because response profiles exhibited strong HG responses at the onset [0–0.4 s], and noticeably weaker HG responses in the following peri-stimulus time-window [0.4–2 s], we analysed these time-windows separately in subsequent analyses. We found that HG onset responses proportionally increased as a function of stimulus energy (Fig. 2a, onset window: 0–0.4 s; $r^2_{[0-0.4s]} = 0.92$; $p < 10^{-4}$) and rapidly decreased thereafter. HG responses did not significantly correlate with salience in the early ($r^2_{[0-0.4s]} = 0.245$,

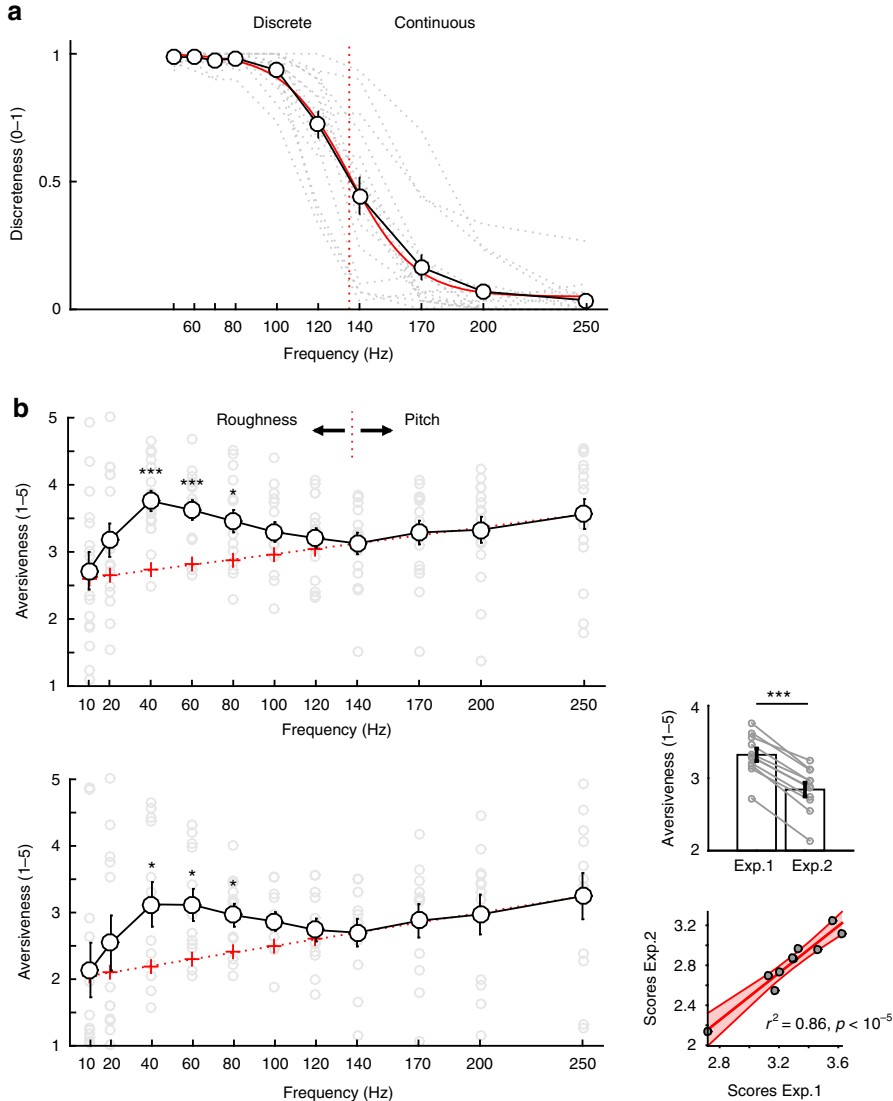

**Fig. 1** Temporal perceptual transition and subjective assessment of temporal salience. **a** Temporal discretisation limit experiment (Experiment 1). Averaged subjective reports of discreteness (1–discrete, 0–continuous) averaged across participants indicate that subjective percept switches from discrete to continuous around 130 Hz (red dotted line: 50% discreteness ratings), which roughly corresponds to the upper limit of the roughness acoustic attribute. The continuous red curve represents the sigmoidal fit to the average across participants. Dotted light grey lines correspond to individual data. **b** Temporal salience experiments. Top panel: Experiment 2a. Averaged subjective aversion (reported on a 1–5 scale) follows a nonlinear pattern. Above the discretisation limit (>130 Hz, i.e., in the pitch range), aversion is linearly proportional to the frequency (energy) of the stimulus. Below this limit, in the roughness range, subjective aversion follows a nonlinear profile and is maximal at 40 Hz. Divergence from linearity in the roughness range is measured as the difference with aversion values predicted by a linear extrapolation of averaged aversion values measured above the discretisation limit. Light grey circles correspond to individual data Bottom panel: Experiment 2b: independent replication of Experiment 2a using the same stimuli at lower intensity (50 dB SPL) in $n = 12$ additional participants. Upper right inset: Main effect of sound intensity between Experiments 2a and 2b. Grey data points correspond to stimuli frequencies. Lower right inset: correlation plot between averaged ratings at each frequency in Experiment 2a and 2b. Error bars indicate SEM. ** and *** indicate significant $p$-values at 0.01 and 0.001, FDR corrected

$p = 0.094$) nor late peri-stimulus periods ($r^2_{[0.8–1.8s]} = 0.352$; $p = 0.058$). In line with the hypothesis that temporally salient sounds should entrain responses in a steady-state, sustained manner with time, we aimed to measure the one to one temporal mapping between the trains of transients (clicks) and brain responses. Therefore, we extended our investigation to a measure that is sensitive to the phase alignment of brain responses at the rate of the exogenous stimulation, the *Cerebro-Acoustic Coherence* (CAC, see "Methods") between sounds and brain responses in the late time window (restricted to [0.8–1.8 s] to avoid potential contamination by onset and offset responses). CAC showed a sustained increase throughout the duration of the sound, and varied

nonlinearly as a function of frequency (Fig. 2a, bottom right). In the late time window, CAC was maximal in the roughness (40–130 Hz) range and qualitatively resembled subjective aversion profiles (see Fig. 1b).

Extending this exploration to 11 patients, we functionally identified auditory electrodes showing a significant HG onset response to the stimuli. Most responsive electrodes lay in the middle and superior temporal regions (Fig. 2b). Averaging responses at these electrodes across patients, we found that early HG activity reflected stimulus energy (Fig. 2b, $r^2_{[0–0.4s]} = 0.6$; $p = 10^{-3}$) but not salience ($r^2_{[0–0.4s]} = 0.2$; $p = 0.137$). We then measured the sustained entrainment (CAC) of brain responses

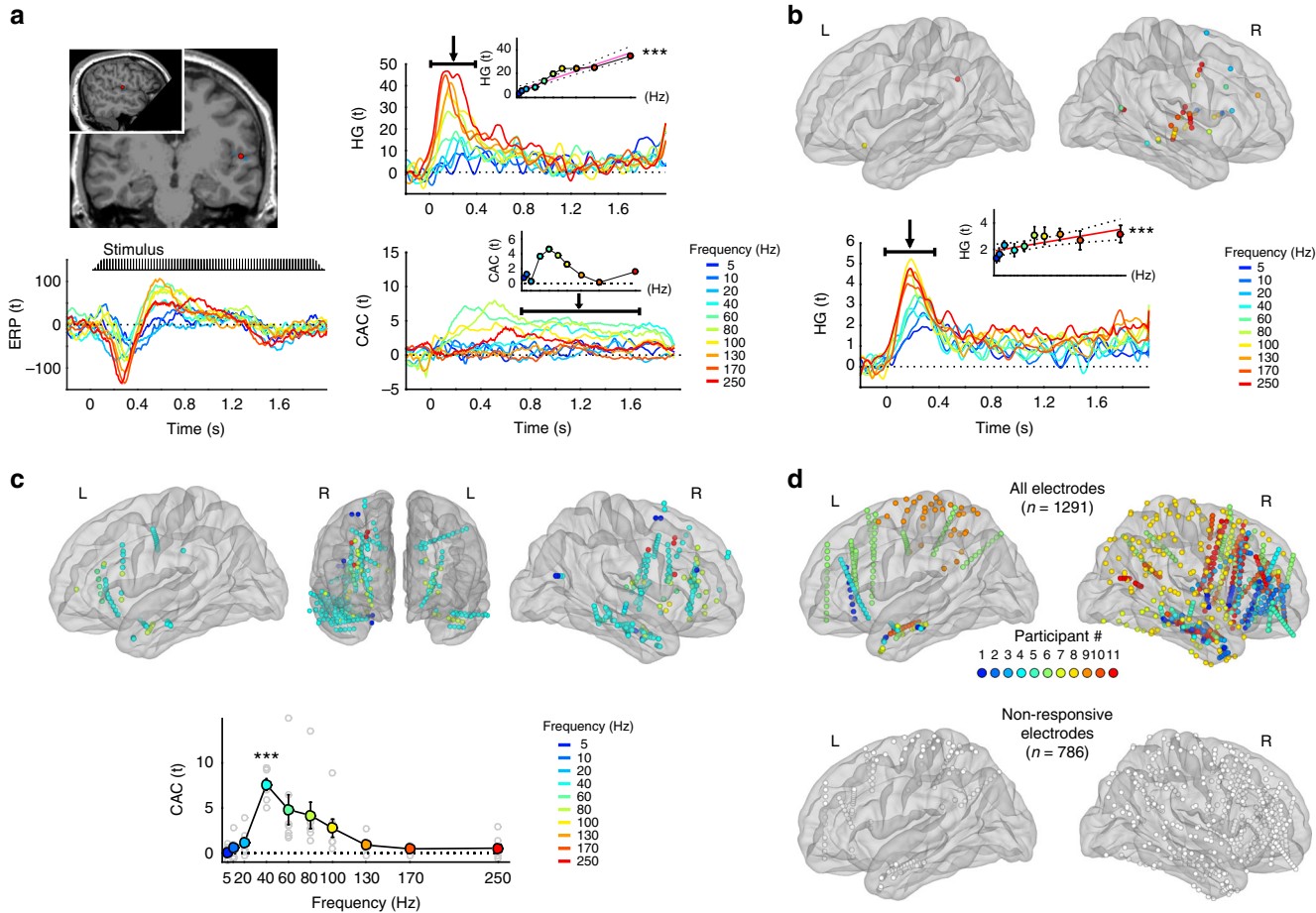

**Fig. 2** HG and CAC spatial response patterns differ across stimulation frequencies. **a** Neural responses to click trains in a representative electrode situated in the right primary auditory cortex (Heschl's Gyrus) of patient S1. Lower left plot: time course of unfiltered event-related responses to the stimuli expressed in t-values relative to baseline. Responses are aligned with an exemplar of a stimulus (here a 40 Hz click train). Upper right plot: time course of high-gamma [HG, 70–200 Hz] amplitude in response to the stimulus, expressed in t-values relative to baseline, as a function of click train frequency (ranging from 5 Hz, blue to 250 Hz, red). HG onset [0–0.4 s] response is linearly proportional to the rate of the stimulus (upper inset). Lower right plot: magnitude squared stimulus-brain coherence (cerebro-acoustic coherence, CAC, expressed in t-values relative to baseline). CAC shows a sustained pattern through the duration of the stimulus that varies nonlinearly as a function of the stimulus rate and is prominent in the roughness range (40–100 Hz; lower inset). **b–d** Neural responses to click trains across all eleven patients. **b** Upper panel: spatial location of electrodes exhibiting significant HG onset responses to sounds. Colours represent the stimulation frequency at which HG response is maximal. Lower plot: time course of HG [70–200 Hz] responses averaged across these electrodes, expressed in t-values relative to baseline, as a function of stimulus frequency. HG amplitude at the response onset [0–400 ms] is linearly proportional to the rate of the stimulus (inset). **c** Electrodes showing sustained, significant stimulus-brain coherence (CAC) are spatially located in widespread cerebral areas. Colours represent the stimulation frequency at which CAC is maximal. Lower plot: CAC as a function of stimulus rate. Light grey circles correspond to individual data. **d**. Top panels: spatial location of all electrodes, colour-coded as a function of participant number. Lower panel: spatial location of electrodes that did not show any significant HG or CAC response. Error bars indicate SEM. Asterisks in panels **a** and **b** indicate significant correlation with stimulus frequency: *** indicates significant p-values at 0.001

by the stimuli during the late time window. Over all frequencies, we found a larger proportion of electrodes exhibiting sustained CAC (24%, Fig. 2c) than HG onset responses (4%, $t_{CAC>HG(10)} = 2.9$; $p = 0.016$). CAC magnitude correlated with the behavioural pattern of subjective aversion ratings identified earlier ($r^2_{[0.8–1.8s]} = 0.358$; $p = 0.031$) but not with stimulus energy ($r^2_{[0.8–1.8s]} = 0.057$, $p = 0.288$). In addition, and in contrast with the more focal concentration of HG onset responses, sustained stimulus-coherent brain responses were anatomically more widespread, involving multiple brain regions (Fig. 2c and Supplementary Movie 1). The spatial extent of this effect (proportion of synced electrodes across patients) differed across stimulus frequencies ($F_{(9, 90)} = 8.59$; $p = 0.003$; Supplementary Movie 1), and correlated with salience ($r^2_{[0.8–1.8s]} = 0.295$; $p = 0.029$) but not stimulus frequency ($r^2_{[0.8–1.8s]} = 0.072$; $p = 0.238$).

**Regional selectivity in the temporal lobe**. Given the apparently distinct spatial patterns of responses in HG and CAC, we first aimed at investigating the regional selectivity of these responses in the temporal lobe, classically involved in rough sounds processing[19,20]. Using an anatomically defined parcellation of temporal sulci and gyri[21], we measured onset HG and sustained CAC profiles in each electrode situated in the temporal lobe (Fig. 3a). This approach first revealed that onset HG responses averaged across anatomical regions were particularly prominent in the electrodes located on the surface of the Superior Temporal Gyrus (Fig. 3b, see also Fig. 2b). Sustained CAC responses, on the other hand, were observed in widespread regions of the temporal lobe (Fig. 3c). Measuring the correlations with either stimulus frequency or perceived aversion (salience), we found that while HG magnitude was proportional to stimulus frequency mainly in

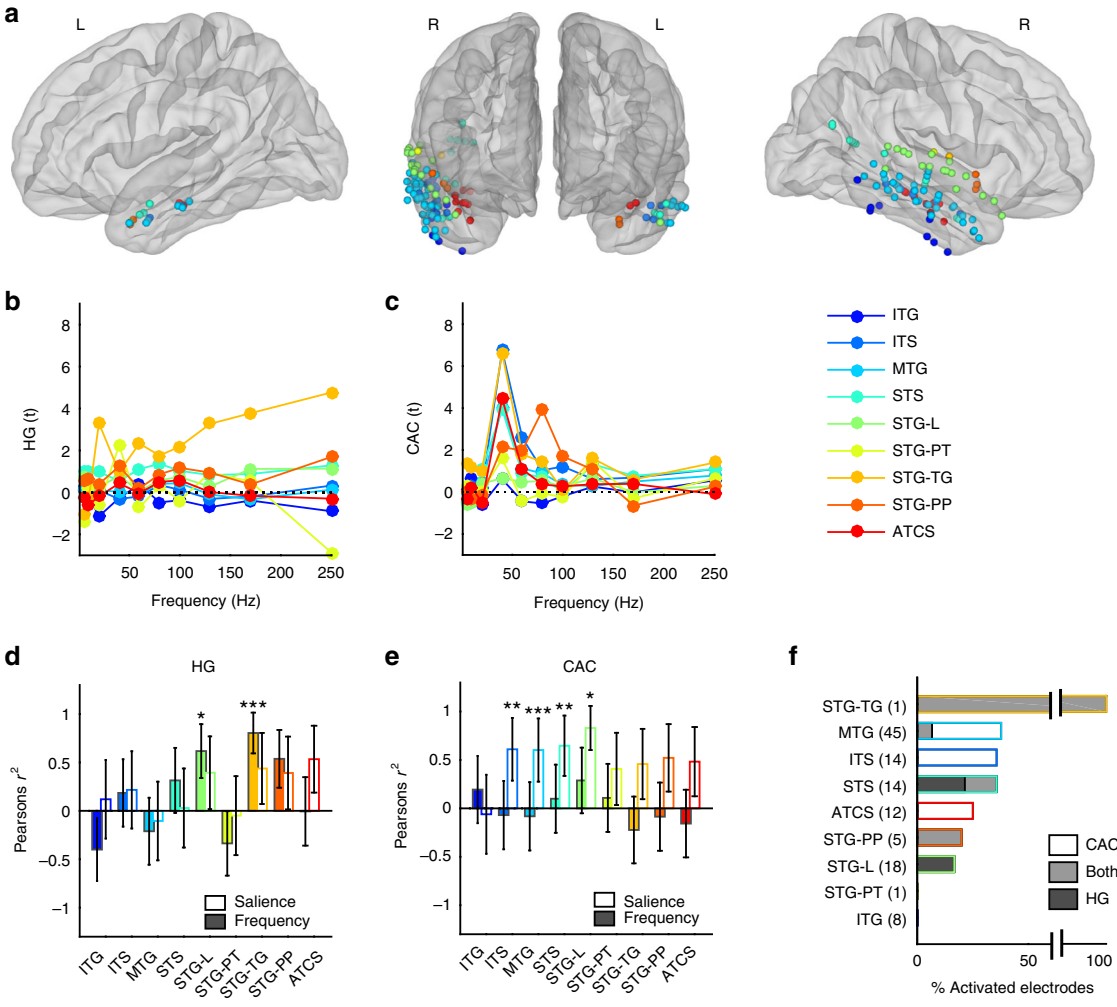

**Fig. 3** HG and CAC response patterns in anatomically defined sub-regions of the temporal lobe. **a** Anatomical and functional categorisation of electrodes in the temporal lobes, based on the Destrieux anatomical parcellation. **b** HG responses in onset [0–0.4 s] window (expressed in *t*-values relative to the baseline) averaged within regions and across participants at each stimulus frequency. **c** Same as in b. for CAC in late [0.8–1.8 s] window. **d** Pearson's correlation value ($r^2$) between onset HG responses and stimulus frequency (coloured filled bars) and salience (empty bars). Error bars indicate SE of the correlation. **e** Same as in (**d**) for CAC. **f** Proportion of activated electrodes exhibiting significant HG onset response (dark shading), sustained CAC (no shading) or both (grey shading) in each network. Numbers on the y-axis indicate the number of electrodes located in each regions. *, ** and *** indicate significant (corrected) *p*-values at 0.05, 0.01 and 0.001, respectively. ITG Inferior Temporal Gyrus, ITS Inferior Temporal Sulcus, MTG Middle Temporal Gyrus, STS Superior Temporal Sulcus, STG-L Superior Temporal Gyrus-Lateral, STG-PT Planum Temporale, STG-TG Transverse Gyrus, STG-PP Planum Polare, ATCS Anterior Transverse Collateral Sulcus

superior temporal regions (Fig. 3d), CAC correlated with salience in temporal regions situated more ventrally to the STG, namely the Inferior Temporal Sulcus, the Middle Temporal Gyrus and the Superior Temporal Sulcus (Fig. 3e, all Pearson's $r^2 > 0.36$, all $p < 0.01$).

Building upon the observation that sustained responses to rough sounds are not confined to superior temporal auditory cortical regions but instead spread to other brain regions, we next sought to expand the anatomical and functional characterisation of these effects to other cerebral areas.

**Regional selectivity in subcortical and limbic regions.** We first investigated whether rough features might additionally recruit regions classically involved in the processing of aversive stimuli, namely key nodes in subcortical and limbic regions[22,23]. To test this, we localised the position of electrodes within anatomically defined subcortical (Amygdala, Hippocampus) and limbic cortical

regions (Insular, Parahippocampal and Entorhinal cortices)[24,25] (Fig. 4a). While these regions exhibited rather weak HG responses (Fig. 4b), CAC was particularly strong in the roughness range in the Hippocampus and Insula (Fig. 4c). HG magnitude in the insula was proportional to both stimulus frequency ($r^2 = 0.58$, $p = 0.02$) and salience ($r^2 = 0.54$, $p = 0.04$, Fig. 4d), whereas the CAC profiles correlated with salience in the hippocampus as well as in the insular, parahippocampal and entorhinal cortices (all $r^2 > 0.34$, all $p < 0.05$, Fig. 4e). Among those regions, the parahippocampal and insular cortices exhibited significant CAC entrainment in more than 25% of all electrodes situated in these regions (Fig. 4f).

**Regional selectivity in parietal and frontal regions.** In addition to synchronising medial temporal and limbic cortical and subcortical regions, one striking aspect of the widespread CAC pattern evidenced at the outset of this work (Fig. 2c) is the apparent

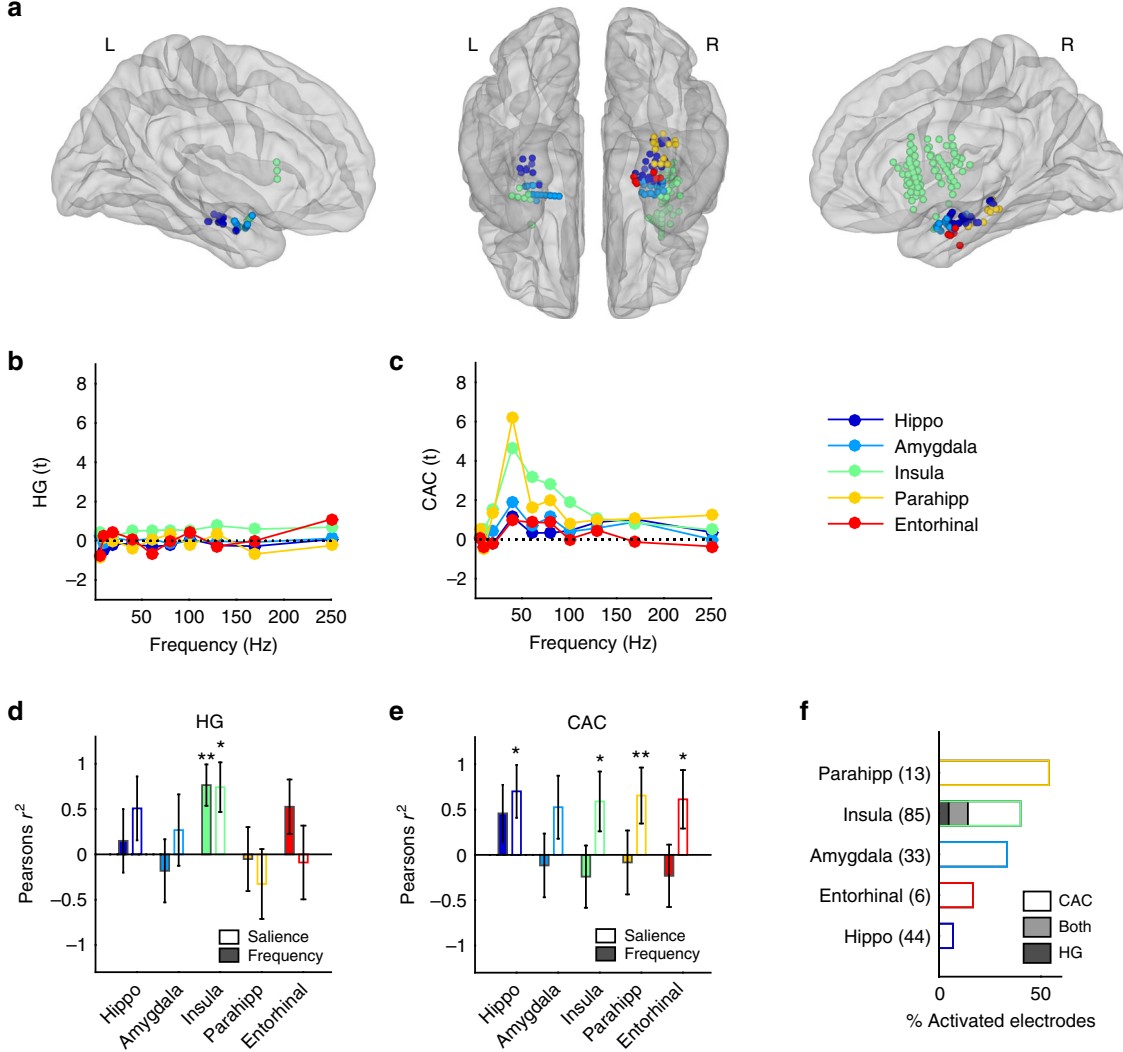

**Fig. 4** HG and CAC response patterns in subcortical and limbic regions. **a** Anatomical and functional categorisation of electrodes in subcortical nuclei and limbic regions, based on the Desikan-Killiani anatomical parcellation. Left: left hemisphere, medial view. Centre: ventral view. Right: right hemisphere, medial view. **b** HG responses (expressed in t-values relative to the baseline) averaged within regions and across participants at each stimulus frequency. **c** Same as in b. for CAC. **d** Pearson correlation value ($r^2$) between onset HG responses and stimulus frequency (colour-filled bars) and salience (empty bars). Error bars indicate SE of the correlation. **e** Same as in (**d**) for CAC. **f** Proportion of activated electrodes exhibiting significant HG onset response (dark shading), sustained CAC (no shading) or both (grey shading) in each region. Numbers on the y-axis indicate the number of electrodes located in each region. *, ** indicate significant (corrected) p-values at 0.05, 0.01, respectively. Hippo Hippocampus, Parahipp Parahippocampal Gyrus

spatial concentration of CAC-activated electrodes in the frontal lobe. Pursuing our exploratory approach, we then sought to measure HG and CAC patterns in the frontal and parietal lobes. Localising the electrodes in anatomically defined cortical regions of the frontal and parietal lobes, we measured HG and CAC responses in each of these regions (Fig. 5a). Again, while HG responses were rather weak in these regions (Fig. 5b), CAC responses exhibited a reliable entrainment profile in the roughness range, as hitherto observed in other regions (Fig. 5c). Measuring how these responses correlate with either stimulus frequency or perceived salience, we found that while HG magnitude was proportional to salience in the caudal Middle Frontal region (Fig. 5d, $r^2 = 0.58$, $p = 0.036$), CAC correlated with salience in most tested frontal regions (Fig. 5e, all significant $r^2 > 0.36$, all $p < 0.05$) as well as in the inferior parietal cortex ($r^2 = 0.79$, $p = 0.0001$). Among frontal regions, those showing the highest percentage of CAC-activated electrodes spanned the

inferior frontal gyrus and orbital frontal cortices, anterior to insular cortex, which we found to be highly recruited by rough sounds earlier (Fig. 4).

**Regional selectivity in resting-state networks**. Given the broad spatial extent of neuronal entrainment to rough sounds that largely exceeds classical auditory regions, we hypothesised that the selective propensity of rough sounds to induce aversive percepts may rely on their capacity to recruit supra-modal (i.e. not specifically auditory) cortical networks. To test this, we localised the position of electrodes within functionally defined networks using an atlas-based parcellation[26] (Fig. 6a). First, we confirmed that onset HG responses mostly activated the sensorimotor (SM) network, which includes the auditory cortex (Fig. 6b) and correlated with stimulus frequency in the SM network only (Fig. 6d). Then, focusing on sustained stimulus-brain coherence (CAC), we found that in all of the seven tested networks (Fig. 6c), coherence

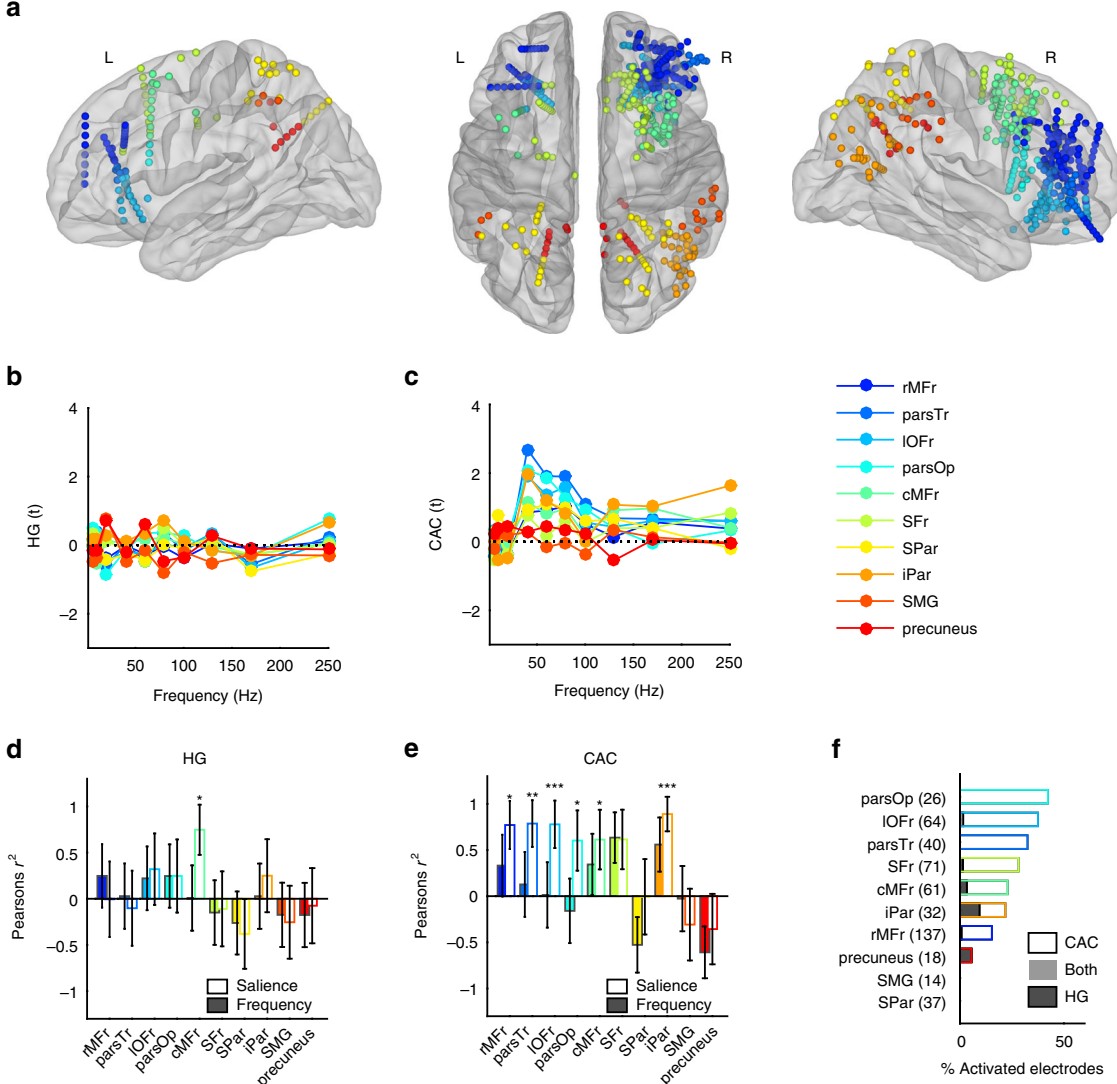

**Fig. 5** HG and CAC response patterns in frontal and parietal sub-regions. **a** Anatomical and functional categorisation of electrodes in frontal and parietal lobes, based on the Desikan-Killiani anatomical parcellation. **b** HG responses (expressed in $t$-values relative to the baseline) averaged within regions and across participants at each stimulus frequency. **c** Same as in (**b**) for CAC. **d** Pearson's correlation value ($r^2$) between onset HG responses and stimulus frequency (colour-filled bars) and salience (empty bars). Error bars indicate SE of the correlation. **e** Same as in (**d**) for CAC. **f** Proportion of activated electrodes exhibiting significant HG onset response (dark shading), sustained CAC (no shading) or both (grey shading) in each network. Numbers on the y-axis indicate the number of electrodes located in each region. *, ** indicate significant (corrected) p-values at 0.05, 0.01, respectively. rMFr rostral Middle Frontal, parsTr Pars Triangularis, iOFr inferior Orbito-Frontal, parsOp Pars Opercularis, cMFr Caudal Middle Frontal, SFr Superior Frontal, SPar Superior Parietal, iPar inferior Parietal, SMG Supra-Marginal Gyrus

profiles significantly correlated with the aversion pattern (Fig. 6e). Comparing the proportion of electrodes showing reliable HG or CAC responses, we found that about 30% of electrodes entrained in a steady-state manner, not only in SM but also in neural networks involved in the modulation of attention and arousal, such as the default-mode (DMN) and the ventral attention (VA) networks (Fig. 6f). Altogether, these results suggest that temporally salient, rough sounds massively synchronise cortical regions, with a possible dominance of the effect in those networks involved in exogenous attention and salience processing.

## Discussion

Our results validate the hypothesis that fast repetitive acoustic transients confer sounds a rough sonic texture and thereby amplify their salience and enhance neural processing. In addition to validating the relevance of the notion of *temporal salience* as a

critical ingredient of exogenous attention, these findings shed new light on how temporal information is coded, processed and perceived by the human brain.

First, assuming that rough auditory percepts are analogous to flickering phenomena in the visual modality, we predicted that subjective aversion should be maximal at frequencies below the transition from roughness to pitch perception. In a series of psychophysical experiments, we demonstrate that temporally enriching sounds enhanced perceived aversion in the roughness range, namely below the transition from discrete to continuous percepts.

The brain is not equally sensitive to all frequencies across the audible spectrum. Since the work of von Helmholtz[5], it has been known that temporal modulations in the roughness range induce more unpleasant subjective percepts than those in adjacent frequency ranges[2,27]. Measuring aversion as a function of click train

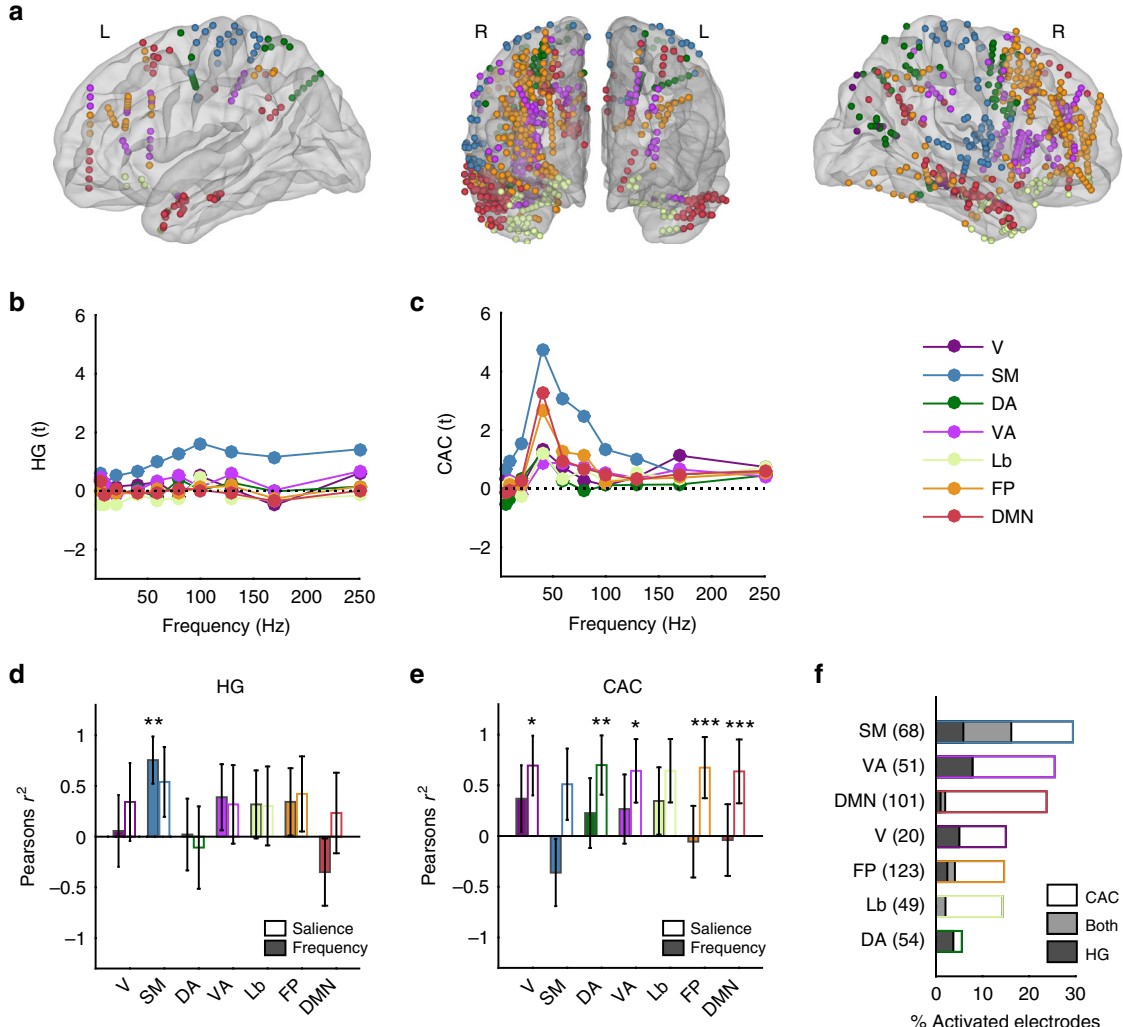

**Fig. 6** HG and CAC response patterns in functional connectivity networks. **a** Anatomical and functional categorisation of electrodes based on functional connectivity networks: sensori-motor (SM); limbic (Lb), default mode (DMN), ventral attention (DA), fronto-parietal (FP) and dorsal attention (DA). **b** HG responses (expressed in $t$-values relative to the baseline) averaged within regions and across participants at each stimulus frequency. **c** Same as in (**b**) for CAC. **d** Pearson's correlation value ($r^2$) between onset HG responses and stimulus frequency (colour-filled bars) or salience (empty bars). Error bars indicate SE of the correlation. **e** Same as in (**d**) for CAC. **f** Proportion of activated electrodes exhibiting significant HG onset responses (dark shading), sustained CAC (no shading) or both (grey shading) in each network. Numbers in parentheses on the $y$-axis indicate the overall number of electrodes located in each network. *, ** and *** indicate significant (corrected) $p$-values at 0.05, 0.01 and 0.001, respectively

frequency, we show that subjective reports linearly follow stimulus frequency above the subjective transition from discrete (rough) to continuous (pitch) percepts (~130 Hz), but not below. Instead, sounds in the roughness range induce highly aversive, buzzing percepts that cannot be accounted for merely by stimulus energy. The phenomenological distinction between roughness and pitch is classically attributed to a dual neural auditory coding strategy of temporal and spectral information[10,14,28]. Adding to this view, our data suggest a new account on the origin of this subjective discrepancy: these two distinct response modes[15,16] may actually reflect the recruitment of two different neural routes for sound processing in the human brain. In one mode, sounds induce a typical onset response in auditory cortical areas, resulting in a transient neuronal activity (HG amplitude) increase that reflects stimulus energy[19,29]. The other mode differs in several aspects. First, it is best measured as a sustained, phase-locked response at the rate of stimulation and is maximal within the roughness (40–80 Hz) range. Second, our intracranial data unequivocally show that responses in this mode synchronise a

large network of supramodal brain areas extending well beyond auditory cortex (Fig. 2c and Supplementary Movie 1), thereby providing decisive evidence in favour of a distributed source of steady-state responses in this range[30]. Indeed, rough sounds massively synchronise widespread cortical regions in a sustained manner, with a relative regional preference for those neural networks that are involved in (or directly affected by) salience detection (VA and DMN[31,32]) over goal-oriented, central-executive networks (DA and FP[33–36]). More specifically, rough sounds not only synchronise cortical auditory regions but also subcortical and cortical limbic as well as frontal regions. Altogether, these findings point to a supramodal neural correlate of temporal salience: the aversive sensation induced by rough sounds results from the persisting, exogenous synchronisation of large-scale networks involved in salience—rather than specifically auditory—processing. This also suggests that the negative percept induced by rough sounds such as dissonant intervals[5], alarm sounds[6] or annoying vocal effects (e.g. vocal fry[37]) might result from their capacity to massively, exogenously hijack brain

networks involved in aversion and pain processing[22]. One alternative—but compatible—interpretation is that such negative percept arises from the difficulty to focus on the task at hand, e.g., interpreting speech.

Neuronal synchronisation in the gamma range has been proposed as a mechanism for selectively routing information and for synchronising activity within and across brain networks[38]. In the context of the current study, one hypothesis that naturally arises is that stimulus-driven, privileged entrainment in the gamma range may reflect the long-range effective recruitment of attentional and arousal related brain regions. In addition to the large extent of spatial synchronisation patterns, it is remarkable that these networks preferentially resonate in a frequency range (30–80 Hz) that tightly matches a well-known endogenous brain rhythm, the so-called gamma band[39]. In the following, we will argue that this correspondence is unlikely to be coincidental.

According to the "communication through coherence" (CTC) hypothesis[38,40,41], gamma oscillations play an instrumental role in enabling communication across distant brain areas: neuronal synchronisation in this frequency band enhances the gain of information transmission, and thereby the effective connectivity between them. In line with the CTC hypothesis, endogenous (top-down) attention, or the voluntary effort to improve one's detection and processing of sensory events, is known to enhance gamma synchronisation in cortical networks[42]. On the other hand, exogenous (bottom-up) attention, which is driven by the characteristics of the sensory stimulus (more specifically its salience) rather than by the cognitive requirements of the task[43], has received less attention in the literature[44]; in particular, whether exogenous attention relies on specific neural circuits and oscillatory mechanisms remains unclear[45]. Introducing the concept of *temporal salience*, we provide a new heuristic to address this issue. Previous work suggested that gamma rhythms might subserve the bottom-up propagation of information in the brain[46,47]. One prediction that naturally ensues is that exogenously entraining neural responses in the gamma range should facilitate neural synchronisation across brain areas and, as a result, enhance percept salience, a prediction that was indeed supported by our results. As a consequence, in addition to validating the paradigmatic pertinence of *temporal salience*, these results provide indirect experimental support to the CTC hypothesis.

Our findings also have interesting implications for the understanding of acoustic communication. Producing salient auditory features to catch the attention of others is a primordial purpose of vocal communication. Here, we validate the hypothesis that temporally enriching sounds—in the *roughness* range—amplifies sensory salience and improves neural and behavioural efficiency. This finding connects with the recent observation that roughness is exploited in natural and artificial alarm signals as a privileged acoustic niche to warn conspecifics[6]. Showing that such sounds recruit salience systems in the human brain and enhance perception, we confirm the fitness of these sounds to ultimately promote the efficient transmission of alarm signals. We further provide evidence in favour of the hypothesis that the use of roughness in alarm signalling is not an epiphenomenon of vocal production. Instead, the selection of communicative features depends on their propensity to induce specific behavioural responses. In this view, the use of roughness in alarm signals reflects an adaptation of communication to the receiver's auditory sampling constraints to hijack her brain, enhance her perception of incoming danger and manipulate her reactions from a distance to ultimately promote survival.

## Methods

**Participants**. These studies received IRB approval by the Commission cantonale d'éthique de la recherche, République et canton de Genève, and all participants gave written informed consent to participate. Twenty-seven healthy participants (15 women, 20–37 years) provided informed consent to participate in the behavioural experiments; they received a monetary compensation for their participation. All participants were right-handed, with normal hearing and no history of neurological disorders. Additionally, 11 patients with drug-resistant focal epilepsy (six women, 18–42 years) undergoing iEEG recordings as part of the workup for epilepsy surgery participated in an auditory experiment. To ensure adequate power, the sample size of behavioural experiments was pre-determined on the basis of prior experimental works assessing subjective ratings of auditory stimuli[6]. The sample size for iEEG recordings was determined to match the preliminary behavioural experiments as well as to ensure sufficient anatomical electrode coverage across participants.

**Stimuli and procedures**. All stimuli were digitally generated using MATLAB with a sampling rate of 96 kHz and presented in a pseudo-randomised order using Psychtoolbox (Version 3.0.12).

The first behavioural experiment (Exp. 1) aimed at determining the *temporal discretisation limit*. Sixteen participants subjectively reported the discreteness of click trains (1 s duration, with 100 ms sine ramping onset and offset; click rise/fall time of 0 ms, plateau time of 1 ms, presented at ~60 dB SPL). Participants used button presses to report whether the sounds were discrete (1) or continuous (0). Click train frequency varied between 50 and 250 Hz. The lowest frequency was selected to avoid that subjective judgments of discreteness are biased by very slow stimuli (<20 Hz). Below this frequency, trains are not perceived as a single stimulus anymore, but rather as a succession of distinct auditory events[10,12,28]. Since we aimed to measure the limit at which one can discretize acoustic transients (clicks) within a stream, we thus intentionally chose a reference value higher that the lowest limit of the roughness range (~30 Hz).

The second behavioural experiment (Exp. 2a) aimed at measuring subjective aversion as a function of the frequency of click trains. The same participants who participated in Exp. 1 also reported the aversion of click trains of varying frequencies on a 5-points Likert-like subjective scale ranging from tolerable (1) to unbearable (5). Click trains (1 s duration, with 100 ms sine ramping onset and offset; click rise/fall time of 0 ms, plateau time of 1 ms, presented at ~60 dB SPL) were presented at frequencies varying between 10 and 250 Hz.

The third behavioural experiment (Exp. 2b) aimed at replicating Exp. 2a at lower sound intensity. Twelve different participants reported the aversion of click trains of varying frequencies on a 5-points Likert-like subjective scale ranging from tolerable (1) to unbearable (5). Click trains (1 s duration, with 100 ms sine ramping onset and offset; click rise/fall time of 0 ms, plateau time of 1 ms, presented at ~50 dB SPL) were presented at frequencies varying between 10 and 250 Hz.

Eleven patients with epilepsy undergoing investigation with iEEG electrodes listened to click trains of varying frequencies and provided a rating of aversion on a 5-points scale. Click trains (2 s duration, with 100 ms sine ramping onset and offset; click rise/fall time of 0 ms, plateau time of 1 ms, presented at ~70 dB SPL) were presented at frequencies varying between 5 and 250 Hz. Patients rated all sounds as highly aversive, regardless of stimulus frequency. This could be due to the need for the sounds to be played at a louder than comfortable volume in order to overcome the naturally noisy environment of the epilepsy monitoring unit, as well as to heightened anxiety caused by the stressful situation the patients found themselves in. Therefore, instead of using patients' own subjective reports of aversion, we correlated their neural data with the subjective values averaged across the 16 normal participants in Experiment 1 (see Fig. 1b).

**Processing and analysis**. In Exp. 1, the discretisation limit was determined using a sigmoidal fitting procedure in the individual discreteness values across frequencies. The discretisation limit was calculated individually by predicting the frequency corresponding to a discreteness of 0.5 and is reported, at the group level, using the mean ± SEM across participants. The goodness of fit (red line in Fig. 1a) is determined by measuring the Pearson correlation between predicted and actual data points (averaged across participants).

In Exp. 2a and 2b to test the linearity (or nonlinearity) of subjective aversion responses as a function of stimulus frequency in the pitch (>130 Hz) and in the roughness (<130 Hz) domains, we used a linear fitting procedure as follows. First, we assessed the linearity of responses in the pitch range by measuring the goodness of fit between predicted and actual responses (averaged across participants). We then aimed to measure how much responses in the roughness range deviated from the extrapolation of the linear prediction below the discretisation limit (red line in Fig. 1b). To do so, we, (1) measured the goodness of fit between predicted and actual values and (2) tested how much actual values (grey dots in Fig. 1b) differed from predicted values (red crosses in Fig. 1b) by applying $t$-tests across participants at each stimulus frequency. In both pitch and roughness windows, goodness of fit is determined by measuring the Pearson correlation between predicted (red crosses) and actual data points (empty black dots, averaged across participants).

Intracranial electrophysiological activity was recorded using linear shafts of depth electrodes (Ad-Tech Medical, Racine, WI, or Dixi Medical, Chaudefontaine, France) implanted by stereotactic surgery ($N = 9$ patients), or grids and strips of subdural electrodes (Ad-Tech Medical, $N = 2$) implanted through a craniotomy, in order to localize the origin of epileptic seizures. Signals were digitised (2048-Hz

sampling rate; Brain Quick LTM, Micromed, Treviso, Italy) and stored to disk for offline analysis.

Intracranial electrodes were localised using the iELVis toolbox (http://ielvis.pbworks.com;[48]). Briefly, we co-registered a high-resolution post-implantation CT scan with a pre-implantation 3-T 3D T1 MRI scan. We then identified iEEG electrodes manually on the CT scan using BioImage Suite 3 (http://www.bioimagesuite.org). We used FreeSurfer (http://surfer.nmr.mgh.harvard.edu) to segment the white matter and deep grey matter structures, reconstruct the pial surface, parcellate the neocortex according to gyral anatomy[24], and register the pial surface of individual patients onto a standardised atlas.

In order to attribute an iEEG electrode to one of the resting functional MRI networks described by Yeo et al.[49], we first brought electrode coordinates from the patient space to the MNI305 template space using an affine transformation (implemented in iELVis). We then coregistered the 'liberal' 7-network parcellation available for the MNI152 brain in FreeSurfer to the MNI305 brain using FSL's FLIRT[50].

In order to label electrodes according to each individual patients' anatomy, we used the Destrieux parcellation[21] for a precise labelling of cortical sulci and gyri in the temporal lobe, and the Desikan-Killiany parcellation[24] for the remaining cortical regions, and the Fischl[25] parcellation for subcortical regions. In order to ensure the reliability and readability of our observations, we do not report effects in those regions targeted by less than ten electrodes across patients (with the exception of the enthorinal cortex, for an exhaustive assessment of temporo-limbic areas). As a consequence, due to the limited number of electrodes available in the cingular cortex ($n < 10$ electrodes across patients in each sub-region of the Desikan-Killiany parcellation of the cingular cortex), we could not reliably assess the response profiles in these regions.

All iEEG data analyses were performed in MATLAB (MathWorks) using the Fieldtrip (http://fieldtrip.fcdonders.nl) package and in-house custom code. Channels displaying excessive epileptiform activity or noise were excluded from the analysis. Line noise was removed using the discrete Fourier transform filtering option in fieldtrip at 50, 100 and 150 Hz. All electrodes were then re-referenced to a common average and visually inspected for electrical artefacts. The filtered, re-referenced, artefact-inspected iEEG data were then epoched using 1 s pre-stimulus to 2 s post-stimulus windows. For each trial, we then subtracted the pre-stimulus baseline activity (from 800 ms before to stimulus onset) in each electrode.

Auditory electrodes were functionally defined as those exhibiting significant high-gamma (HG) responses in the 400 ms following sound onset (significance threshold for HG 'activation' was set at $p < 0.01$ after FDR correction across all electrodes). This window was determined based on visual inspection of the data, to dissociate early onset effects to later sustained entrainment (see below). High-gamma activity, which reflects local neuronal activity[18], was defined as the mean normalised amplitude envelope of frequencies between 70 and 200 Hz. HG amplitude time series were computed by band-pass filtering the iEEG signal in 10 Hz bands from 70 to 200 Hz. The envelope of each narrow band signal was obtained by taking the absolute value of the analytic signal obtained from a Hilbert transform. Amplitude time series were then amplitude normalised, averaged and multiplied by the mean amplitude across all bands. This normalisation procedure aims to correct for the $1/f$ decay in the EEG power spectrum. These processing steps resulted in a single broadband amplitude time series representing a proxy for local neuronal activity[18] recorded at each electrode contact. We then transformed these data into time-courses of $t$-scores (calculated at each time point across trials) per contact, condition and participant. Overall, data were distributed normally, which allowed us to use standard parametric tests (e.g., paired $t$-test, repeated-measures ANOVA) to assess the statistical significance of observed effects.

In accordance with our hypothesis that temporal salience should be reflected in the entrainment of neural responses to click trains, we sought neural responses that, (1) were sustained across the peri-stimulus time-course and (2) correlated with subjective aversion reports. As no sustained HG response was reliably observed across electrodes, we extended our investigation to a measure that is sensitive to the phase alignment of brain responses at the rate of the exogenous stimulation. To do so, we measured the coherence between the stimulus waveform and the preprocessed brain signals (cerebro-acoustic coherence, CAC) at the stimulus frequency.

In order to best model the relationship between the stimulus and the brain response while avoid spurious effects potentially due to the coherence measure we used, we first transformed the waveform to model the peripheral transformation of the sound occurring at the cochlear level. Sound waveforms were transformed into a time-frequency representation (spectrogram) using a filter-bank approach. Waveforms were filtered using 128 different linear-phase finite impulse response filters (512th order). Filters were designed to estimate critical bands[51] with centre frequencies logarithmically spanning the frequency space and corresponding to equivalent rectangular bandwidths (according to the equation: $BW = 24.7 * (F * 4.37 + 1)$, where BW denotes bandwidth and $F$ denotes a center frequency in KHz[52]). Each filter's output was then Hilbert transformed in order to extract the analytic amplitude and log-transformed. We then averaged the output of this filter-bank processing to provide a novel waveform that better reflect the output of the cochlea[51]. Note that although this approach was meant to realistically model the processes occurring at the cochlear level, not applying this transformation did not significantly change the results.

CAC was obtained by measuring the Magnitude Squared Coherence Estimate using the MATLAB function mscohere.m. This function calculates an estimate of the magnitude-squared coherence between the input (the sound waveform) and the output (the brain signal in the time window of interest) using Welch's averaged, modified periodogram method. Coherence is a function of frequency with values between 0 and 1 that indicate how well the input corresponds to the output at a considered frequency. The magnitude squared coherence, $C_{io}$, is given by $C_{io} = |P_{io}|^2/(P_{ii} \times P_{oo})$, where $P_{ii}$ and $P_{oo}$ are the power spectral density (PSD) estimate of the input $i$ and the output $o$, respectively; and $P_{io}$ is the Cross-PSD estimate of $i$ and o. CAC was calculated in each electrode, individual and condition. In order to avoid a potential contamination of entrained responses to sound onsets, we focused our analysis on a later time window—defined by visual inspection of the data—ranging from 800 to 1800 ms after stimulus onset. CAC values were baseline-corrected by subtracting a surrogate measurement of CAC between stimulus and brain activity during a 1000-ms period preceding sound onset to capture endogenous activity at the stimulus frequency. We then transformed these data into $t$-scores (calculated per time-window across trials) per contact, condition and participant. Using this approach, we could then identify subsets of significantly "entrained/sustained" electrodes for each stimulus frequency (significance threshold for CAC 'activation' was set at $p < 0.01$ after FDR correction across all electrodes). In order to qualitatively show how CAC evolves in time as a function of stimulus frequency for illustrative purposes (Fig. 2a, bottom panel), we also applied this method in a single electrode using a sliding 1000-ms window in 10-ms steps across the whole peristimulus time-course.

Of note, we initially intended to use the CAC approach[53] to measure the temporal relationship between neural responses and click trains waveform while taking into account the shape of these stimuli. However, because of the peculiar spectral content of clicks and despite the use of a cochlear filter bank to model the output of the cochlea (see previous section), it remained possible that the CAC method might not be best suited to capture these effects and may introduce unwanted confounds in the analysis. Therefore, we aimed to reproduce our basic CAC finding (Fig. 2c, bottom panel) using an alternative, frequency-tagging approach that does not depend on the stimulus shape. Importantly, all other analysis steps (e.g. baseline correction) were the same as those described earlier for CAC. Here, instead of measuring stimulus-brain coherence (CAC), we measured the power of neural responses filtered at the stimulus frequency. As for the extraction of HG activity, amplitude time series were computed by band-pass filtering the iEEG signal, but this time, at the stimulus frequency $F \pm 0.5$ Hz. The envelope of each narrow band signal was obtained by taking the absolute value of the analytic signal resulting from a Hilbert transform. As this method appeared to provide slightly less robust SNRs, significance threshold for frequency-tagging 'activation' was set at $p < 0.05$ (instead of $p < 0.01$ for CAC) after FDR correction across all electrodes. We then obtained a similar graph of $t$-values across frequencies and participants (Supplementary Fig. 1a) as the one we previously obtained for CAC (Fig. 2c, bottom panel). Measuring correlations between the outcomes of each CAC and Frequency-tagging approaches on averaged (Supplementary Fig. 1b) and individual (Supplementary Fig. 1c) data, we demonstrated that the two approaches yielded very similar results, thereby validating the use of CAC measurements to robustly measure sustained steady-state responses.

All statistical tests were corrected for multiple comparisons using FDR correction[54] unless otherwise stated.

In order to assess the strength of HG and CAC measurements across participants within clusters of electrodes of interest, two distinct approaches were taken. For electrode subsets defined functionally (i.e. on the basis of functional activation, in Fig. 2b and c), individually $t$-scored HG or CAC data were averaged within subsets of electrodes at the individual level, before statistical testing at the group level. For anatomically defined electrode subsets (i.e. on the basis of atlas-based parcellations, in Figs. 3–6), we had to use a different averaging strategy because in most cases electrodes from only a few patients were available per anatomical region of interest. Therefore, we did not perform analyses at the group level, but instead we pooled data from all available electrodes in the considered subset of interest. Our aim here was to quantify the putative relationship between activation profiles (namely HG or CAC across stimuli frequencies) and stimulus energy or perceptual salience. To do so, we measured Pearson's correlations between these response profiles (averaged across all the electrodes of the considered subset) and either stimulus energy or subjective aversion reports. Pearson's correlation analyses were subjected to a specific statistical correction approach. Because correlations with stimulus frequency or salience often involved fewer than 10 data points, we aimed to correct for potential biases using a non-parametric approach. The non-parametric correlation statistic was performed by repeating 5000 times the calculation of a permutation test where the experimental conditions (stimulus frequency or salience) are randomly intermixed before measuring the correlation with neural data. Finally, we calculated the corrected p-values by comparing the values of the statistics of our original data with the statistics of all permutations.

## Data availability

Data are available upon reasonable request from Luc H. Arnal (luc.arnal@unige.ch).

## Code availability

Custom made code is available at https://github.com/LucArnal/SoundOfSalience.

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

## Acknowledgements

We thank S. Marchesotti and S. Martin for comments on the manuscript and the staff of the HUG Epilepsy unit. P.M. is supported by Swiss National Science Foundation grant # 167836.

## Author contributions

L.H.A. designed the experiments, performed the research and analysed the data. P.M. contributed to data analysis. P.M. and L.S. assisted with iEEG recordings. L.H.A., A.K., A.L.G. and P.M. wrote the manuscript. Correspondence and requests for materials should be addressed to L.H.A.

## Additional information

**Competing interests:** The authors declare no competing interests.

