## [Transparent Peer Review File · Nature Communications]

Reviewers' comments:

Reviewer #1 (Remarks to the Author):

See below.

Reviewer #2 (Remarks to the Author):

The paper presents a two-part study concerning temporal salience of auditory stimuli, the first a behavioral study in healthy volunteers, and the second taking advantage of intracranial electrophysiology from epilepsy neurosurgery patients in an attempt to delineate the relevant brain networks. The design of the experiment is compelling and the analyses suggest many interesting findings that fit together to explain what auditory properties encourage temporal salience as well as the brain responses/networks that mediate them. In particular, the manuscript suggests an attractive idea that the particular unpleasantness of click repetitions or amplitude modulations of ~40 Hz arise due to stimulating brain networks that operate at the same frequency. Intracranial EEG is certainly a powerful technique for revealing auditory networks and their connectivity at gamma frequencies. However, I have some concerns about a few key aspects of the paper (perhaps misunderstandings on my part) that I would like the authors to clarify.

The detection efficiency results are reported in the text to peak at 170 Hz, but Figure 1A apparently shows a peak around 100 Hz. Is there a typo here or have I misunderstood the connection with the graph?

The sound localization results also don't appear to be so clear-cut to me. The U-shaped profile that is claimed appears to be driven only by the 500 Hz result. One can imagine that if the 500 Hz result were an outlier, this U-shaped profile would no longer be evident.

The cerebro-acoustic coherence measure is applied between the ECoG data and stimulus waveforms consisting of click trains. However, coherence measures are usually applied between two "rich" signals, e.g., EEG and EMG, or perhaps EEG and naturalistic sounds like phonemes. Click trains have an inherently peculiar power spectrum with broadband components arising from the sharpness of the clicks in addition to other components relating to the frequency of the clicks. Is the coherence of brain signals with such a waveform meaningful, or could it arise merely due to the spectral properties of the click train?

For example, why should there be the same low coherence at ≤ 20 Hz as there is above ≥ 130 Hz? The authors suggest that clicks start to fuse into a continuous percept around 130 Hz, but click frequencies ~20 Hz are very clearly perceptible, quite rapid... and as I have just synthesized such a click train myself, I can attest they are still rather annoying. The authors point out that the network activated by 40 Hz click trains are much more widespread than with 20 Hz: if this is true, I would suspect 20 Hz to CAC of a magnitude similar to 40 Hz, but perhaps more restricted to primary auditory areas. It would be helpful to show if this were the case.

Perhaps more confidence in this analysis could be supported by testing coherence of the click trains with synthesized ECoG-like signals, or by shuffling the relationship between the real ECoG data and the corresponding click trains. Maybe the technique of spike-field coherence could be adapted to this situation as an alternative?

The authors should clarify whether the intracranial EEG electrode data were collapsed across subjects prior to statistical comparison, or if statistics were performed across subjects. Otherwise, the statistical procedures for the behavioral experiments appear sound, and correction for multiple comparisons with FDR is appropriate for the performed analyses.

Minor issues:

The definition of "efficiency" was not straightforward to understand; it would be appreciated to include the equation directly in the methods section of this manuscript. Instead, the manuscript refers to an earlier paper by the first author, but even there, its definition is only found in the supplementary material.

The source code for the analyses has not been made available, as is required by Nature's policy. This would be especially helpful not only reproducing results but also to examine details of how important measures in the manuscript (like CAC) were computed.

Figure 1a/1b: it seems that the y-axis should simply be labelled z-score? Only the circles represent efficiency, but RT and accuracy appear on the same plot.

abstract: "but instead" -> "but also" ?

typo on page 7: "intracacial"

Reviewer #3 (Remarks to the Author):

Arnal et al. report a series of behavioral experiments and human intracranial EEG recordings that address the perceptual benefits of temporally salient auditory stimuli and the neural mechanisms underlying such benefits. They find that subjective unpleasantness, coherence between the acoustic signal and brain response (CAC), and coherence between different brain areas peak for 40Hz click train stimuli. Based on this result, Arnal et al. suggest that exogenous entrainment of intrinsic gamma band oscillations across sensory and attentional networks induces perceptual salience as well as subjective unpleasantness. This is interpreted in support of the Communication-through-coherence (CTC) hypothesis, which states that communication across brain areas relies on neuronal synchronization in the gamma frequency band.

The paper reports four behavioral experiments, which measure the effect of temporal modulation rate (5 - 400 Hz) in tones and click trains onto stimulus detection in noise (Experiment 1), left/right stimulus localization (Experiment 2 for tones and Exp. 3 for clicks), and localization at different loudness levels (Experiment 4). Two additional experiments (5 and 6) tested the threshold at which click trains are perceived as a continuous stimulus (temporal sampling limit experiment) and showed that this threshold is intimately related to ratings of subjective unpleasantness of click trains (subj. unpleasantness experiment). Finally, an iEEG experiment explored the effects of temporal modulation rate (10 - 200 Hz) on neural responses to click trains.

I have several major concerns regarding the connection between the 7 experiments reported in this manuscript that undermine the interpretability of the results. Moreover, some statistical procedures need to be described in more details, and others appear inappropriate and might have lead to inflated significance (Exp 1- 4, listed further below).

The first concern is about the relationship between temporal salience, subjective pleasantness, and roughness/continuity perception. Temporal salience is operationalized in Experiments 1- 4 through participants' perceptual performance (detection/localization) for stimuli with different modulation rates. These experiments show that performance is best within some, experiment-specific, modulation range. This range depends on the experiment, but in all four studies the optimal modulation frequency lies in the range of 100- 200 Hz. On the contrary, Experiments 5 and 6 show that unpleasantness peaks at 40 Hz in the roughness/discontinuity range. It remains unclear how unpleasantness relates to the salience measures introduced in Experiments 1-4. If both measure

the same thing, why are best frequencies so different between Exp1-4 and Exp5-6? If they measure different processes - as suggested in the discussion section - how is unpleasantness related to temporal salience and what is the relevance of the iEEG experiment to the concepts introduced in Exp 1-4? Overall the relation between experiments 1-4 and experiments 5-7 remains unclear.

The second major concern is the importance of the 40Hz frequency band for temporal salience. Unpleasantness, CAC, and CCC clearly peak for 40 Hz click trains. However, it remains unclear how this relates to temporal salience and processing efficiency, which were characterized in Experiments 1-4 and peaked at much higher frequencies, above 100 Hz and outside the gamma range. An alternative hypothesis would be that signal processing in noise or at low intensities, as tested in Exp. 1-4, actually corresponds to evoked responses to stimulus onset, which monotonically increased with modulation frequency (Fig. 3a+b). Unfortunately the iEEG experiment did not include modulation frequencies >400 Hz to test this hypothesis. It remains unclear how sound detection and localization relate to 40 Hz oscillations or the gamma band in general.

Overall this paper seems to contain two separate data sets: 1) Results on the temporal sampling limit in different perceptual tasks (Experiment 1- 4). 2) Neural signatures of subjective unpleasantness at different temporal modulation rates (Experiment 5-7).

Detailed comments

Introduction

- P.2 2nd paragraph of introduction: This paper does not address the neural basis of the transition from roughness to pitch percept, as misleadingly suggested by this paragraph.

- P.2: What is a strobe phenomenon? Here, and similarly in the discussion, the authors refer to findings from vision research that might not be known to the target audience of this paper. Please explain analogies to visual processing.

Results & Figures

Experiments 1 to 4

Statistical analysis of linear vs. polynomial effects (i.e. inverted u-shape) in behavioral analysis: The authors calculate the correlation between actual and predicted behavioral measures using a linear and a quadratic model. They then test for a difference between those correlations. Because the linear model is nested in the quadratic model, the correlation value for the quadratic model will always be at least as good as for the linear model.

The t-tests reported for the quadratic model fit in the results are thus trivial and do not indicate whether the quadratic fit reflects an improvement over the linear fit.

A correct test of whether the quadratic term improves the model fit will take into account the degrees of freedom of each model, which are smaller for the quadratic model.

One way would be to report significance values for the regression coefficients for the linear and quadratic terms in the full (i.e. quadratic model). If the quadratic fit is better than the linear fit, the coefficient for the quadratic term will be significantly <0 (due to the inverted u-shape) across participants. The model could either be fit across participants with a linear mixed-effects approach, or using a hierarchical approach, whereby the within-participant beta-values are t-tested against 0 at the group level (similar to the group analysis approach taken in fMRI).

Without knowing the results of this analysis, I'm afraid that the results of the behavioral experiments 1-4 remain hard to interpret.

1) To identify the range of modulation frequencies that lead to best behavioral performance, it is necessary to identify the frequencies at which performance drops significantly below the peak performance level. Please add the corresponding pairwise comparisons to the results section.

2) The exact best performance modulation frequencies are only explicitly stated in the discussion section, not in the results section. Please add them to the results section (backed by appropriate

quantifications, see above).

Experiments 5-6:

Methods section does not provide details on fitting and statistical testing procedure for the sigmoidal curve of discreteness/continuity perception.

Again, please provide quantitative backup for the maximal unpleasantness range of 40-60 Hz.

Experiment 7 - iEEG recordings

I have a series of concerns regarding the statistical analyses of the iEEG data. Without these additional details it is hard to interpret the results.

Page 7:

- 1) The temporal range used in analysis of evoked responses in HG is very broad (0-400), especially given the excellent temporal resolution of ecog. How does narrowing it down or using only a window around the peak alter the results?
- 2) No power is reported for analysis of coherence in different frequency ranges. Please report whether power in the relevant frequency bands was increased and discuss them with respect to the other results.
- 3) Coherence analysis: Please explicitly explain in the methods section why coherence values are obtained timepoint by timepoint, rather than a single value per patient, electrode, and condition across the time course of the analysis window. Overall the description of the coherence metric is hard to follow.
- 4) For example electrode 1 as well as for the group analysis, the maximal coherence is indicated in the 40-130 Hz range. However in Figure 3, the peak in coherence seems to be at 60Hz, whereas coherence appears at almost 0 at 130 Hz. Please indicate significance against 0 at different coherence levels if the exact range of stimulus frequencies at which coherence is maximal is important.
- 5) Please indicate the overall number of electrodes in this analysis.

Page 9 (Cortico-cortical coherence):

- 1) The frequency range of 30-80 Hz is very broad (>1 octave), and it is questionable whether oscillatory entrainment happens throughout this range or whether it reflects short bursts of activity outside the narrow band of 40-50 Hz, in which evidence for oscillatory processes is ample (e.g. in work by P. Fries).
- 2) What is the evidence/argument that favors communication between brain areas as source of coherence over coincidence based on the same rhythmic input? An analysis that could alleviate this concern partial out the exogenous effects of the stimulus before testing for cortico-cortical coherence.

P18: 'We then transformed these data into time-courses of t-scores (calculated at each time point across trials) per contact, condition and participant. As a result, data were distributed normally, which allowed us to use standard parametric tests (e.g., paired t-test, repeated-measures ANOVA) to assess the statistical significance of observed effects.'

It is not clear what the authors mean here. If they are describing a linear normalization such as z-scoring (which seems to be the procedure used here, despite the name 't-scores'), z-scoring does not change the shape of the distribution, e.g. from skewed to normal. Rather, z-scoring only changes the range of values to allow for comparability across baselines. If they are referring to some non-linear transformation that changes the shape of data distribution to resemble a normal distribution, this needs to be described in more detail.

Minor comments:

P3: "We hypothesize that providing fast, but still perceptible, acoustic temporal cues". Likely means fast but still perceptible as distinct inputs and not a continuous stream. The wording is ambiguous.

P. 3: What is the roughness range? Please define and use consistently throughout the paper.

P.4: 'Because of neuronal (or peripheral) adaptation mechanisms, perceptual efficiency is presumably only optimal at the onset of sensory transients and later becomes suboptimal, especially at low SNRs, when the risk of missing the sound onset are higher' please provide a reference to the literature to back up this claim.

Figures:

Figure 1:

Please add axes for accuracy and reaction time in panels a and b.

Please change x-axis in panels a and b to scale. The categorical scale distorts the scale of the frequency effects and might induce a wrong percept of the curvature of the effects.

Also, it would be helpful to indicate (e.g. with vertical lines) the range in which performance is at best and does not differ between stimulus frequencies.

Figure 3:

Panel b: Caption indicated that electrodes are localized to auditory areas, however, electrodes are also marked in medial temporal (not really auditory anymore) and prefrontal areas. please report % electrodes in each area that responded to quantify this claim. From caption it is not clear whether this plot shows electrodes pooled across all patients in this study or only portions of the data. Please specify.

Panels a-c: Modulation frequency range appears to be 5-250 Hz, but methods section indicated 10-200. Please check and correct.

Panel c: if little black dots indicate outliers, they are hardly visible. Also, they are not mentioned in caption.

Figure 4: The use of the term CAC in figure 4 is confusing because it's cortico-cortical and not cortico-acoustic coherence. Is this average coherence across time points?

Please specify.

Reviewer #1 (Remarks to the Author):

This paper combines psychoacoustic tests and iEEG to demonstrate the behavioral, subjective and neural salience of auditory roughness. It's principal contribution is that click trains at rates below the auditory flutter fusion threshold drive neural activity over large portions of cortex, leading to widespread long-range synchrony. As such, it provides the most compelling neurobiological account of auditory roughness to date and is thus highly relevant to the study of auditory perception and auditory-vocal communication. The paper is well written and the methods are well described. My recommendation is for acceptance after minor revision.

Overarching issues

My main issues with the paper are: (1) the idea that the auditory flutter fusion threshold can be equated with "the sampling capacity of the auditory system"; and (2) the complexity of the transformations applied to the neural data in order to make inferences about cortical activation and neural synchronization. Point 1 should be relatively simple to omit or defend. Point 2 can be addressed by further explanation and clarification.

POINT 1. On page 4 3rd para / page 5 1st para, the authors argue that the results of their detection & localization experiments show that the facilitory effects of rough sounds are limited by "the sampling capacity of the auditory system", going on in the following paragraph to describe an experiment aimed at determining the auditory flutter fusion threshold, which they interpret as representing the "sampling limit" of the auditory system.

It is not clear whether the authors are using this DSP concept literally or metaphorically. A literal interpretation does not seem justified as any conception of a global sampling limit in the auditory system would be mired in the highly context sensitive nature of auditory perception, as well as variation across measurement techniques (e.g., Besser, 1967). A metaphorical interpretation would apply but is arguably more misleading than informative.

Besser GM (1967) Some physiological characteristics of auditory flutter fusion in man. *Nature*, 214:17-19.

I recommend replacing all instances of this assumption with more direct descriptions of the actual results. For example, "This suggests that the facilitory effect of temporally enriching sounds is limited by the sampling capacity of the auditory system" -> "This suggests that the facilitory effect of temporally enriching sounds is frequency limited".

The point of this criticism is to make the links between (1) the facilitory effects of sub auditory flutter threshold stimulation, (2) the transition from discrete to continuous percepts, and (3) the perception of roughness *more explicit*. In other words, to more directly connect the peaks in curves 1a and 1b with the pre-transition zone of the sigmoid in 2a, and the significantly aversive frequencies in 2b. This would stand on its own. There is no need for a vague reference to the sampling rate of the auditory system.

POINT 2. The most intriguing results of the paper critical depend on understanding the "cerebo-acoustic coherence (CAC)" measure (also referred to as "stimulus-brain coherence"). Currently, the only explanation in the main text is that it represents "the coherence between sound and brain responses" as the "magnitude squared coherence estimate at the stimulus rate". Devoting a sentence or two (of the main text) to describing what the authors take this measure to reflect would greatly improve the clarity of the manuscript, especially for readers unfamiliar with coherence analyses. (see also comments below on the methods). The CAC is interpreted to

reflect “entrainment” and “phase-locking” between neural activity and the stimuli. The added text should specifically justify these claims.

Minor points

Please provide line numbers in any further revisions.

INTRODUCTION

Page 2, last para - “*temporally salient*” - either italicization or quotes will suffice. After the initial occurrence it is not necessary to continue italicizing this term.

Page 3, 1st para - typo: psychoacoustics -> psychoacoustic

Page 3, 1st para - (*i*) italicization unnecessary

RESULTS - *Behavioural assessment of temporally salient acoustic regimes*

Page 3, 3rd para - The depth of amplitude modulation is not specified here or in the methods. This is an important factor in the perception of roughness (Vassilakis, 2001) and should be described.

Vassilakis, P. N., [Perceptual and Physical Properties of Amplitude Fluctuation and their Musical Significance], Doctoral Dissertation, University of California, Los Angeles, Los Angeles (2001).

Page 3, 3rd para - “In other words, adding fast temporal cues within the roughness range (but not beyond 170 Hz) improved participants’ ability to detect target sounds presented at a low SNR.” - Estimates of the range of frequencies at which roughness is perceived vary across studies and are dependent on multiple factors (e.g., carrier frequency and modulation depth). Add references to clarify the approximate range used here.

Page 3, 3rd para - “We tested the ability of participants to localize a pure 700 Hz tone modulated at varying AM frequencies (ranging from 0 –no modulation– to 500 Hz, **Fig. 1b**) that were lateralized to their left or right” - The fact that the localization stimuli were also presented in noise is not currently specified in the main text description of this experiment and should be added.

Page 3, 3rd para - “to limit unwanted additional spectral cues” - The relevance of the tone/click train distinction vis-a-vis roughness will not be clear to non-specialists. Nor will be the reason for the switch here. An example of a specific cue that switching to clicks is meant to avoid would be useful, as would a brief explanation of why this matters.

Page 4, 2nd para - “neuronal (or peripheral)” - Distinction not clear. Replace “neuronal” with “central” if that is what is meant.

Page 4, 2nd para - The fact that the click trains were also presented in noise needs to be added somewhere. Also, consider specifying the noise intensity in the legend for Figure 1c, or relabeling the x-axis as SNR.

RESULTS - *Subjective characterization of temporal salience*

Page 5, 2nd para - “the perceptual transition from discrete to continuous –the temporal sampling limit– sits at 130 ± 6.5 Hz.” see POINT 1 above.

Page 5, 3rd para - Typo: “at frequencies close below the temporal sampling limit”

Page 5, 3rd para - Consider providing a sentence and/or reference to justify the fact that rough sounds are perceived as unpleasant/aversive. Otherwise the response scale, which only allows aversive responses, could appear biased (even though the responses indicate that it is appropriate).

Page 5, 4th para - “the roughness range” again. This term is problematic without some explanation up-front and/or references.

Page 5, 4th para - “the ability of clicks trains to exogenously induce sustained entrainment of responses in time.” Typo: “clicks” -> “click”. Also, “exogenously” is superfluous; click trains cannot be endogenous.

Page 6, figure 2 legend - “which roughly corresponds to the upper limit of the roughness acoustic attribute” -reference?

RESULTS - *Neurophysiological correlates of temporal salience*

Page 7, 1st para - “early auditory cortex” Is there are reason that more standard terminology is avoided here (e.g., primary or core auditory cortex)?

Page 7, 3rd para - “Then, focusing on sustained stimulus-brain coherence, we found that over 60% of electrodes entrained in a steady-state manner, not only in SM but also in neural networks involved in the modulation of attention and arousal, such as the default-mode (DMN), limbic (Lb) and ventral attention (VA) networks” - Consider adding “60% of [all electrodes recorded from] entrained...”. Also, in figure 3d plotting the locations of electrodes that did not entrain would help clarify. Finally, how was classification of an electrode as “entraining in a steady-state manner” made? Was a CAC threshold used? Please specify. Consider rephrasing: “... entrained in a steady-state manner, meaning that _____. These electrodes were located not only in SM...”

Page 7, 3rd para - “six tested networks”? What does “tested” mean here? Consider: “examined”

Page 8, figure 3a - specify stimulus onset and offset times graphically or in legend text.

Page 8, figure 3b legend - “neural response in the high-gamma range are localized in auditory regions”. Add qualifying statement and/or numbers. Many of the electrodes in 3b appear to be located outside traditional “auditory” regions.

Page 8, figure 3d legend - Typo? Should ref 14 be ref 15?

Why have DFs for the t-tests have been omitted throughout?

RESULTS - *A privileged regime for efficient neuronal communication-through-coherence*

Page 10, figure 4a legend - "(average proportion of synchronized electrode pairs)" - How was the classification of "synchronized" made?

Are the terms "subjective salience" (figure 4a legend), "aversiveness pattern" (p7, 4th para), and "subjective unpleasantness" (p9, 1st para) all used to refer to the same thing? If so, sticking to one term would clarify.

Page 10, figure 4b legend - What is the difference between the bottom left panel ("connectivity pattern") and the 40 Hz "long-range synchronisation pattern" in the top row supposed to show? Please specify. Also, the meaning of line color is explained, but the meaning of line thickness is not (are they redundant?). Finally, at the end, consider adding "based on Desikan-Killiany parcellation" to the end of the sentence "Bottom right panel: connectivity pattern at 40 Hz across anatomical areas".

DISCUSSION - Temporal saturation of perception and behavioural facilitation

page 11, 2nd para - "ecological context of alarm signaling [in noise and/or] from a distance". Consider adding. Noisy contexts seem just as relevant as distant contexts to the authors' hypothesis.

DISCUSSION - Dual neural routing versus dual neural coding

page 11, 3rd para - "Since the work of von Helmholtz, it [has been] known"

page 11, 3rd para - "...temporal modulations in the roughness range induce more unpleasant subjective percepts than auditory attributes in adjacent frequency ranges" - Auditory attributes is vague. Consider replacing with "temporal modulations". Also, see earlier comment re roughness range.

Page 12, 1st para - "Thereby providing the best neural account of the aversive sensation induced by temporally salient sounds". Fantastic. A simple alternative hypothesis to the one offered at the end of this paragraph is that the negative percepts associated with the massive hijacking of brain networks results from their being distracted from the task at hand, such as interpreting speech.

METHODS

Page 16, 4th para - "We replicated the previous experiment using repetitive sequences of clicks" Add a description of the clicks, e.g., square wave, rise/fall time (= 0s?), plateau time (=0.01s?).

Page 18, 2nd para - "7 networks"? The main text says 6, only 5 of which are named.

Page 18, 4th para - "Amplitude time series were then amplitude normalized, averaged and multiplied by the mean amplitude across all bands. This normalization procedure aims to correct for the 1/f decay in the EEG power spectrum." What does "amplitude normalized" mean here? Why is compensating for the 1/f decay important? This would seem to have the effect of boosting power in the higher frequency gamma bands beyond what was actually observed. Why this was done?

Page 18, 4th para - Replace “ECoG” with “iEEG” for consistency

Page 18, 4th para - “These processing steps resulted in a single broadband amplitude time series representing the mean neuronal activity recorded in each electrode contact.” -> “... representing the mean neuronal activity [between 70-200 Hz] recorded in each electrode contact.” Correct?

Page 18, 5th para - “between the stimulus waveform and the raw brain signals.” - Consider replacing “raw brain signals” with the more specific “raw electrical activity recorded at each electrode”. Also, is “raw” really appropriate here? No preprocessing?

Page 18, 5th para - “Coherence is a function of frequency with values between 0 and 1 that indicate how well the input corresponds to the output at a considered frequency”. Given the central importance of this concept to this work, some further detail and/or an example would be useful here. This detail should elaborate on what “correspondence” means in this context? E.g., total energy at the same frequency independent of phase or lag. Also, briefly explain why the magnitude squared coherence was selected over coherence.

Page 19, 1st para - “In order to avoid a potential contamination of entrained responses to sound onsets, we focused our analysis on a later time window ranging from 800 ms to 1,800 ms after stimulus onset.” Can the authors provide some justification for this particular time window? Why would waiting 800 ms prevent the observation of onset entrained responses? For me, it is fine if this window was based on visual inspection of the data, but the logic should be specified.

Page 19, 1st para - “CAC values were baseline-corrected by subtracting a surrogate measurement of CAC between stimulus and brain activity during a 1000-ms period preceding sound onset”. Confusing because there was no stimulus before sound onset. Consider adding something like “... preceding sound onset, [to capture endogenous activity at the stimulus frequency]”

Rebuttal: Ms. No. NCOMMS-18-08959-T

(L.H. Arnal, A. Kleinschmidt, L. Spinelli, A.-L. Giraud, P. Mégevand)

General comments:

We thank the three reviewers for their positive and thoughtful comments that have helped us reshape and sharpen our arguments. At the outset of our rebuttal, we would like to provide some general comments that are relevant to the various concerns of the reviewers.

- In the initial submission of our manuscript, we reported the results of seven behavioural experiments and one neurophysiological experiment using intracranial recordings in five epileptic patients. All reviewers' comments were quite positive, in particular with regards to the second part of the manuscript. However, it appeared to us that although it reflected our incremental experimental approach, the amount and diversity of behavioural experiments obscured our general message, to the point that the reader could hardly follow the argument that we initially aimed to develop.

One central issue raised by all reviewers pertained to the apparent inconsistency between the first set of behavioural experiments –Exp. 1 to 4 focusing on the processing of temporal cues at low SNR– and the second set of experiments –Exp. 5 to 7 investigating the subjective perception and neural processing of rough sounds–. Although we originally considered this –low vs. high SNR– behavioural dichotomy of potential interest, we acknowledge that interpreting all results under the same notion, the temporal salience, was actually detrimental to the understanding of our main message. Indeed, the two sets of experimental tasks were too different from each other and can hardly be conflated within a single explanatory framework.

In line with all reviewers' comments, we have decided to refocus the manuscript on the most important studies and results, namely those concerning the behavioural and neural underpinnings of aversion to rough sounds, while removing the first four experiments. This decision also aligns with Reviewer 3's critique about the lack of neurophysiological data to backup the claims based on this set of experiments. The manuscript now describes only the behavioural experiments that directly speak to the iEEG data.

- We have added recordings from 6 patients (now the manuscript reports results from eleven patients in total) thereby allowing us to assess brain responses to roughness more reliably and with a much higher anatomical and functional granularity. Although this decision has considerably delayed the resubmission of this manuscript, we arbitrated that the opportunity to record 6 more patients in a few month to consolidate the reliability and generalizability our findings was worth delaying the resubmission. We sincerely hope that both the editors and reviewers will agree with our decision.

- We have changed all the figures, statistical analyses and descriptions according to the reviewers' critiques and the journal's requirements. In line with Reviewer 2's comments about her/his personal assessment of unpleasantness of 20

Hz sounds, and to validate that the pattern of aversiveness rating did not depend on sound intensity, we have replicated this behavioural experiment in twelve subjects at lower intensity (10 dB lower than the first experiment). The results clearly confirm our initial observations and show that the non-linear subjective pattern is indeed a function of stimuli rate but not intensity.

- Finally, in light of relevant and fair theoretical critiques –and because it did not add any critical insights regarding our conclusions–, we have decided to remove the connectivity analysis previously described in the last figure of our previous manuscript. This does not significantly impact the overall interest of our findings.

Overall, the changes made considerably improve the clarity of the manuscript while preserving the main theoretical and experimental appeal of this work, namely the discovery that the aversion to rough sounds is best accounted for by a non-classical, neurally widespread auditory response scheme.

Please find our point-by-point reply to the reviewers below.

Reviewers' comments:

Reviewer #1 (Remarks to the Author):

This paper combines psychoacoustic tests and iEEG to demonstrate the behavioral, subjective and neural salience of auditory roughness. Its principal contribution is that click trains at rates below the auditory flutter fusion threshold drive neural activity over large portions of cortex, leading to widespread long-range synchrony. As such, it provides the most compelling neurobiological account of auditory roughness to date and is thus highly relevant to the study of auditory perception and auditory-vocal communication. The paper is well written and the methods are well described. My recommendation is for acceptance after minor revision.

We thank the reviewer for her/his positive and thoughtful comments and also for the very detailed and careful assessment of our work.

Please see our general responses above and our detailed point-by-point responses below.

Overarching issues

My main issues with the paper are: (1) the idea that the auditory flutter fusion threshold can be equated with “the sampling capacity of the auditory system”; and (2) the complexity of the transformations applied to the neural data in order to make inferences about cortical activation and neural synchronization. Point 1 should be relatively simple to omit or defend. Point 2 can be addressed by further explanation and clarification.

POINT 1. On page 4 3rd para / page 5 1st para, the authors argue that the results of their detection & localization experiments show that the facilitatory effects of rough

sounds are limited by “the sampling capacity of the auditory system”, going on in the following paragraph to describe an experiment aimed at determining the auditory flutter fusion threshold, which they interpret as representing the “sampling limit” of the auditory system.

It is not clear whether the authors are using this DSP concept literally or metaphorically. A literal interpretation does not seem justified as any conception of a global sampling limit in the auditory system would be mired in the highly context sensitive nature of auditory perception, as well as variation across measurement techniques (e.g., Besser, 1967). A metaphorical interpretation would apply but is arguably more misleading than informative.

Besser GM (1967) Some physiological characteristics of auditory flutter fusion in man. *Nature*, 214:17-19.

I recommend replacing all instances of this assumption with more direct descriptions of the actual results. For example, “This suggests that the facilitory effect of temporally enriching sounds is limited by the sampling capacity of the auditory system” -> “This suggests that the facilitory effect of temporally enriching sounds is frequency limited”.

The point of this criticism is to make the links between (1) the facilitory effects of sub auditory flutter threshold stimulation, (2) the transition from discrete to continuous percepts, and (3) the perception of roughness *more explicit*. In other words, to more directly connect the peaks in curves 1a and 1b with the pre-transition zone of the sigmoid in 2a, and the significantly aversive frequencies in 2b. This would stand on its own. There is no need for a vague reference to the sampling rate of the auditory system.

We agree that our –metaphorical– interpretation was confusing. We have replaced all instances of this assumption with more direct references to the perceptual task at play. In accordance with the actual instruction of the behavioural task –namely to determine whether the sound sounds ‘discrete’ versus ‘continuous’– we now systematically refer to the subjective, perceptual transition between roughness and pitch as the “subjective discretization limit”.

We also thank the reviewer for pointing out the useful reference by Besser (1967), that we now cite head on in the introduction section, together with a series of other relevant references (e.g. Helmholtz 1863, Terhardt 1974, Langner 1992, etc) that allow us clarifying the concept of roughness and its subjective perceptual underpinnings (please see below).

POINT 2. The most intriguing results of the paper critical depend on understanding the “cerebo- acoustic coherence (CAC)” measure (also referred to as “stimulus-brain coherence”). Currently, the only explanation in the main text is that it represents “the coherence between sound and brain responses” as the “magnitude squared coherence estimate at the stimulus rate”. Devoting a sentence or two (of the main text) to describing what the authors take this measure to reflect would greatly improve the clarity of the manuscript, especially for readers unfamiliar with coherence analyses. (see also comments below on the methods). The CAC is interpreted to

reflect “entrainment” and “phase-locking” between neural activity and the stimuli. The added text should specifically justify these claims.

Thank you for this comment. Although the notion of stimulus-brain locking is often informally referred to as entrainment or phase-locking in the auditory steady-state response (ASSR) literature, we agree that is important to provide a clearer definition in the context of the current study. We do not intend to make specific claims or assumptions with regards to the neurophysiological mechanism –e.g. oscillatory entrainment– at play here. Rather, we basically aim to measure the one-to-one temporal mapping between stimuli trains and brain responses. As a consequence, we now basically refer to classical terms such phase-locking and steady-state responses.

This has been clarified in the main text as follows:

P4 line 104: “*In line with the hypothesis that temporally salient sounds should entrain responses in a steady-state, sustained manner with time, we aimed to measure the one to one temporal mapping between the trains of transients (clicks) and brain responses. Therefore, we extended our investigation to a measure that is sensitive to the phase alignment of brain responses at the rate of the exogenous stimulation, the Cerebro-Acoustic Coherence (CAC, see Methods) between sounds and brain responses in the late time window (restricted to [0.8–1.8 s] to avoid potential contamination by onset and offset responses).*”

Minor points

Please provide line numbers in any further revisions.

We apologise for this unfortunate omission. Line numbers are now provided in this new version of the manuscript.

INTRODUCTION

Page 2, last para - “*temporally salient*” - either italicization or quotes will suffice. After the initial occurrence it is not necessary to continue italicizing this term.

Corrected.

Page 3, 1st para - typo: psychoacoustics -> psychoacoustic

Page 3, 1st para - (*i*) italicization unnecessary

Corrected.

RESULTS - Behavioural assessment of temporally salient acoustic regimes

We thank the reviewer for these useful remarks. Please note that this set of experiments is not anymore reported in the new version of the manuscript, which now focuses on the subjective characterization of temporal salience using click trains. However, we will take into account and carefully address these relevant comments, should these experiments be described in another publication.

Page 3, 3rd para - The depth of amplitude modulation is not specified here or in the methods. This is an important factor in the perception of roughness (Vassilakis, 2001) and should be described.

Vassilakis, P. N., [Perceptual and Physical Properties of Amplitude Fluctuation and their Musical Significance], Doctoral Dissertation, University of California, Los Angeles, Los Angeles (2001).

Agreed. Please see above.

Page 3, 3rd para - “In other words, adding fast temporal cues within the roughness range (but not beyond 170 Hz) improved participants’ ability to detect target sounds presented at a low SNR.” - Estimates of the range of frequencies at which roughness is perceived vary across studies and are dependent on multiple factors (e.g., carrier frequency and modulation depth). Add references to clarify the approximate range used here.

Please see above.

“ adding fast temporal cues within the roughness range (from ~30 Hz to ~150 Hz)

Page 3, 3rd para - “We tested the ability of participants to localize a pure 700 Hz tone modulated at varying AM frequencies (ranging from 0 –no modulation– to 500 Hz, **Fig. 1b**) that were lateralized to their left or right” - The fact that the localization stimuli were also presented in noise is not currently specified in the main text description of this experiment and should be added.

Please see above.

Page 3, 3rd para - “to limit unwanted additional spectral cues” - The relevance of the tone/click train distinction vis-a-vis roughness will not be clear to non-specialists. Nor will be the reason for the switch here. An example of a specific cue that switching to clicks is meant to avoid would be useful, as would a brief explanation of why this matters.

Please see above.

Page 4, 2nd para - “neuronal (or peripheral)” - Distinction not clear. Replace “neuronal” with “central” if that is what is meant.

Please see above.

Page 4, 2nd para - The fact that the click trains were also presented in noise needs to be added somewhere. Also, consider specifying the noise intensity in the legend for Figure 1c, or relabeling the x-axis as SNR.

Please see above.

RESULTS - *Subjective characterization of temporal salience*

Page 5, 2nd para - “the perceptual transition from discrete to continuous –the temporal sampling limit– sits at 130 ± 6.5 Hz.” see POINT 1 above.

Corrected. We replaced this statement (P3, Line 61) by: “the subjective transition from rough to continuous –the temporal discretization limit– sits at 130 ± 6.5 Hz.”

Page 5, 3rd para - Typo: “at frequencies close below the temporal sampling limit”

Corrected.

Page 5, 3rd para - Consider providing a sentence and/or reference to justify the fact that rough sounds are perceived as unpleasant/aversive. Otherwise the response scale, which only allows aversive responses, could appear biased (even though the responses indicate that it is appropriate).

Fair enough. Thank you for pointing this out. We have added the following sentence:

P2 line 37: *“Fast repetitive modulations produce “temporally salient” flickering percepts (e.g. strobe lights, vibrators, and alarm sounds (Arnal et al., 2015), which efficiently capture attention, generally induce rough and unpleasant sensations, and elicit avoidance (Helmholtz, 1863) ”.*

Page 5, 4th para - “the roughness range” again. This term is problematic without some explanation up-front and/or references.

Agreed. We now provide a brief explanation (and supporting references) in the introduction as follows:

P1 Line 24: *“Another important emerging feature is roughness, an acoustic texture that arises from fast repetitive acoustic transients. Although the delimitation of the roughness range –whether psychoacoustic or perceptual– may slightly vary depending on experimental settings (Besser, 1967; Fastl and Zwicker, 2007; Krumbholz et al., 2000), empirical observations consistently suggest that sensory systems and perception are exceedingly well tuned to recurring temporal features in the 30–150 Hz range (Arnal et al., 2015; Helmholtz, 1863; Langner, 1992; Terhardt, 1974) “*

Page 5, 4th para - “the ability of clicks trains to exogenously induce sustained entrainment of responses in time.” Typo: “clicks” -> “click”. Also, “exogenously” is superfluous; click trains cannot be endogenous.

Fair enough. This has been corrected.

Page 6, figure 2 legend - “which roughly corresponds to the upper limit of the roughness acoustic attribute” -reference?

This has been removed from the legend.

RESULTS - *Neurophysiological correlates of temporal salience*

Page 7, 1st para - “early auditory cortex” Is there are reason that more standard terminology is avoided here (e.g., primary or core auditory cortex)?

We now provide a more precise terminology based on the Destrieux Atlas (Destrieux et al., 2010). Please see also responses to other revs.

Page 7, 3rd para - “Then, focusing on sustained stimulus-brain coherence, we found that over 60% of electrodes entrained in a steady-state manner, not only in SM but also in neural networks involved in the modulation of attention and arousal, such as the default-mode (DMN), limbic (Lb) and ventral attention (VA) networks” - Consider adding “60% of [all electrodes recorded from] entrained...”. Also, in figure 3d plotting the locations of electrodes that did not entrain would help clarify. Finally, how was

classification of an electrode as “entraining in a steady-state manner” made? Was a CAC threshold used? Please specify. Consider rephrasing: “... entrained in a steady-state manner, meaning that _____. These electrodes were located not only in SM...”

Thank you for these useful remarks. The text and figures have been changed accordingly.

Fig. 3-6 (panels f) now report the requested information as follows: “Proportion of activated electrodes exhibiting significant HG onset response (dark shading), sustained CAC (no shading) or both (grey shading) in each network.”.

CAC thresholding is now explained in the online methods section (P4, Line 160): “we could then identify subsets of significantly “entrained/sustained” electrodes for each stimulus frequency (significance threshold for CAC ‘activation’ was set at $p < 0.01$ after FDR correction across all electrodes)”.

Please note that Fig. 2 panel d now depicts the spatial location of all electrodes that did not show any significant HG or CAC response.

Page 7, 3rd para - “six tested networks”? What does “tested” mean here? Consider: “examined”

This sentence is no longer relevant and has been removed.

Page 8, figure 3a - specify stimulus onset and offset times graphically or in legend text.

This information is now provided in relevant panels and legend of Figure 2.

Page 8, figure 3b legend - “neural response in the high-gamma range are localized in auditory regions”. Add qualifying statement and/or numbers. Many of the electrodes in 3b appear to be located outside traditional “auditory” regions.

We have addressed this important issue by providing a better-informed assessment of the anatomical localization of all electrodes in the temporal lobe, according to the anatomically-based Destrieux atlas (Destrieux et al., 2010).

Page 8, figure 3d legend - Typo? Should ref 14 be ref 15? Why have DFs for the t-tests have been omitted throughout?

Thanks for pointing this out. Corrected.

RESULTS - A privileged regime for efficient neuronal communication-through-coherence

In light of relevant and fair theoretical critiques and as this analysis did not really provide any additional information as compared to previous ones, we have decided to remove this approach from the manuscript. This does not significantly impact our conclusions or the overall interest of our findings.

Page 10, figure 4a legend - “(average proportion of synchronized electrode pairs)” - How was the classification of “synchronized” made?

Sentence removed.

Are the terms “subjective salience” (figure 4a legend), “aversiveness pattern” (p7, 4th para), and “subjective unpleasantness” (p9, 1st para) all used to refer to the same thing? If so, sticking to one term would clarify.

Thank you for pointing this out. We now systematically use the terms “aversiveness/aversion” but not anymore “unpleasantness”. On the other hand we intentionally aimed to make a distinction between aversiveness (the subjective sensation, as reported by the participants in the subjective task) from the more general concept of (temporal) salience. First, this allows interpreting our findings regarding sounds discretizability and aversiveness within a common framework. Second, it allows setting our conclusions in line with a general literature about salience processing and exogenous attention, which contains but not necessarily requires aversive emotion or negative sensations. We have clarified that we use aversiveness as behavioural proxy to salience in the main text, which should help the reader better adopting our reasoning.

Page 10, figure 4b legend - What is the difference between the bottom left panel (“connectivity pattern”) and the 40 Hz “long-range synchronisation pattern” in the top row supposed to show? Please specify. Also, the meaning of line color is explained, but the meaning of line thickness is not (are they redundant?). Finally, at the end, consider adding “based on Desikan-Killiany parcellation” to the end of the sentence “Bottom right panel: connectivity pattern at 40 Hz across anatomical areas”.

These analysis and related figure have been removed (please see above).

DISCUSSION - *Temporal saturation of perception and behavioural facilitation*

page 11, 2nd para - “ecological context of alarm signaling [in noise and/or] from a distance”. Consider adding. Noisy contexts seem just as relevant as distant contexts to the authors’ hypothesis.

Suggestion added.

DISCUSSION - *Dual neural routing versus dual neural coding* page 11, 3rd para - “Since the work of von Helmholtz, it [has been] known”

Corrected.

page 11, 3rd para - “...temporal modulations in the roughness range induce more unpleasant subjective percepts than auditory attributes in adjacent frequency ranges” - Auditory attributes is vague. Consider replacing with “temporal modulations”. Also, see earlier comment re roughness range.

Fair enough. Corrected.

Page 12, 1st para - “Thereby providing the best neural account of the aversive sensation induced by temporally salient sounds”. Fantastic. A simple alternative hypothesis to the one offered at the end of this paragraph is that the negative percepts associated with the massive hijacking of brain networks results from their being distracted from the task at hand, such as interpreting speech.

Fair enough. Suggestion added as follows (P7, Line 251):

“This also suggests that the negative percept induced by rough sounds such as dissonant intervals (Helmholtz, 1863), alarm sounds (Arnal et al., 2015) or obnoxious vocal effects (e.g. vocal fry (Oliveira et al., 2016)) might result from their capacity to massively, exogenously hijack brain networks involved in aversion and pain processing (Hayes and Northoff, 2012). One alternative –but compatible– interpretation is that such negative percept arises from the difficulty to focus on the task at hand, e.g. interpreting speech. “

METHODS

Page 16, 4th para - “We replicated the previous experiment using repetitive sequences of clicks” Add a description of the clicks, e.g., square wave, rise/fall time (= 0s?), plateau time (=0.01s?).

Information added.

Page 18, 2nd para - “7 networks”? The main text says 6, only 5 of which are named.

Corrected.

Page 18, 4th para - “Amplitude time series were then amplitude normalized, averaged and multiplied by the mean amplitude across all bands. This normalization procedure aims to correct for the 1/f decay in the EEG power spectrum.” What does “amplitude normalized” mean here? Why is compensating for the 1/f decay important? This would seem to have the effect of boosting power in the higher frequency gamma bands beyond what was actually observed. Why this was done?

We used this method in accordance with a common high-gamma band normalization practice in previous electrophysiological research as described in the following reference: (Fisch et al., 2009).

Page 18, 4th para - Replace “ECoG” with “iEEG” for consistency

Corrected.

Page 18, 4th para - “These processing steps resulted in a single broadband amplitude time series representing the mean neuronal activity recorded in each electrode contact.” -> “... representing the mean neuronal activity [between 70-200 Hz] recorded in each electrode contact.” Correct?

We clarified this in the text follows: (Online Methods P3, Line 113)

“High-gamma activity, which reflects local neuronal activity (Ray et al., 2008), was defined as the mean normalized amplitude envelope of frequencies between 70 and 200 Hz. (...) These processing steps resulted in a single broadband amplitude time series representing the mean neuronal activity (Ray et al., 2008) recorded in each electrode contact.”

Page 18, 5th para - “between the stimulus waveform and the raw brain signals.” - Consider replacing “raw brain signals” with the more specific “raw electrical activity recorded at each electrode”. Also, is “raw” really appropriate here? No preprocessing?

We corrected as follows: *“between the stimulus waveform and the preprocessed*

brain signals.”

Page 18, 5th para - “Coherence is a function of frequency with values between 0 and 1 that indicate how well the input corresponds to the output at a considered frequency”. Given the central importance of this concept to this work, some further detail and/or an example would be useful here. This detail should elaborate on what “correspondence” means in this context? E.g., total energy at the same frequency independent of phase or lag. Also, briefly explain why the magnitude squared coherence was selected over coherence.

Coherence is a quantification of the similarity in frequency content of two signals as a function of frequency. In other words it allows identifying significant frequency-domain correlation at the sine wave frequencies.

Magnitude-squared coherence is similar to coherence (see e.g. <https://www.mathworks.com/examples/signal/mw/signal-ex53096804-coherence-function>), this has been clarified in the text. There exist various implementations of coherence measurements. We have chosen this function because it is readily implemented and available in Matlab. We now provide the corresponding equations in the methods section (P4 line149) for better clarity and reproducibility of the results. We have also published the main scripts of our analyses on: <https://github.com/LucArnal/SoundOfSaliency>.

Page 19, 1st para - “In order to avoid a potential contamination of entrained responses to sound onsets, we focused our analysis on a later time window ranging from 800 ms to 1,800 ms after stimulus onset.” Can the authors provide some justification for this particular time window? Why would waiting 800 ms prevent the observation of onset entrained responses? For me, it is fine if this window was based on visual inspection of the data, but the logic should be specified.

Fair enough. We have clarified this in the main text as follows:

P4: Lines 99-111: *“Because response profiles exhibited strong HG responses at the onset [0-0.4 s], and noticeably weaker HG responses in the following peri-stimulus time-window [0.4–2 s], we analysed these time-windows separately in subsequent analyses. We found that HG onset responses proportionally increased as a function of stimulus energy (Fig. 3a, onset window: 0–0.4s; $r^2_{[0-0.4s]} = 0.92$; $p < 10^{-4}$) and rapidly decreased thereafter. HG responses did not significantly correlate with saliency in the early ($r^2_{[0-0.4s]} = 0.245$, $p = 0.094$) nor late peri-stimulus periods ($r^2_{[0.8-1.8s]} = 0.352$; $p = 0.058$). In line with the hypothesis that temporally salient sounds should entrain responses in a steady-state, sustained manner with time, we aimed to measure the one to one temporal mapping between the trains of transients (clicks) and brain responses. Therefore, we extended our investigation to a measure that is sensitive to the phase alignment of brain responses at the rate of the exogenous stimulation, the Cerebro-Acoustic Coherence (CAC, see Methods) between sounds and brain responses in the late time window (restricted to [0.8–1.8 s] to avoid potential contamination by onset and offset responses).”*

P4 Lines 15: *“In order to avoid a potential contamination of entrained responses to sound onsets, we focused our analysis on a later time window –defined by visual*

inspection of the data– ranging from 800 ms to 1,800 ms after stimulus onset.”

Page 19, 1st para - “CAC values were baseline-corrected by subtracting a surrogate measurement of CAC between stimulus and brain activity during a 1000-ms period preceding sound onset”. Confusing because there was no stimulus before sound onset. Consider adding something like “... preceding sound onset, [to capture endogenous activity at the stimulus frequency]”

Thank you for this suggestion. We have changed the text accordingly.

Reviewer #2 (Remarks to the Author):

The paper presents a two-part study concerning temporal salience of auditory stimuli, the first a behavioral study in healthy volunteers, and the second taking advantage of intracranial electrophysiology from epilepsy neurosurgery patients in an attempt to delineate the relevant brain networks. The design of the experiment is compelling and the analyses suggest many interesting findings that fit together to explain what auditory properties encourage temporal salience as well as the brain responses/networks that mediate them. In particular, the manuscript suggests an attractive idea that the particular unpleasantness of click repetitions or amplitude modulations of ~40 Hz arise due to stimulating brain networks that operate at the same frequency. Intracranial EEG is certainly a powerful technique for revealing auditory networks and their connectivity at gamma frequencies. However, I have some concerns about a few key aspects of the paper (perhaps misunderstandings on my part) that I would like the authors to clarify.

We thank the reviewer for her/his positive and thoughtful comments and also for the very detailed and careful assessment of our work.

Please see our general responses in the introduction of this rebuttal and our detailed point-by-point responses below.

Please note also that in light of the three reviewers comments, for the sake of clarity of the revised manuscript and for the reader to better appreciate the links between behavioural and neural iEEG data, we have decided not to report anymore the first three behavioural experiments in the new version of the ms. We have made a series of substantial changes and additional analyses to improve our new manuscript, which now focuses on the behavioural and neural characterization of the aversiveness of temporally salient sounds. We are very grateful for these comments anyway and for the efforts made to assess this part of our work. Of course, we will take all these constructive critiques into account, should these results be reported in another publication.

The detection efficiency results are reported in the text to peak at 170 Hz, but Figure 1A apparently shows a peak around 100 Hz. Is there a typo here or have I misunderstood the connection with the graph?

Correct. Please note that this experiment is not reported anymore in the new version of the manuscript.

The sound localization results also don't appear to be so clear-cut to me. The U-shaped profile that is claimed appears to be driven only by the 500 Hz result. One can imagine that if the 500 Hz result were an outlier, this U-shaped profile would no longer be evident.

We understand the reviewer concern. For the record, the choice of frequencies reflected our incremental approach in which we aimed at determining the frequency of transition drop in a dichotomous manner. We have replicated and refined the U-shaped pattern using various types of sound patterns (AM tones, clicks) to ensure that this is not spurious.

At any rate, please note that this experiment is not reported anymore in the new version of the manuscript.

The cerebro-acoustic coherence measure is applied between the ECoG data and stimulus waveforms consisting of click trains. However, coherence measures are usually applied between two "rich" signals, e.g., EEG and EMG, or perhaps EEG and naturalistic sounds like phonemes. Click trains have an inherently peculiar power spectrum with broadband components arising from the sharpness of the clicks in addition to other components relating to the frequency of the clicks. Is the coherence of brain signals with such a waveform meaningful, or could it arise merely due to the spectral properties of the click train?

Thank you for this comment. As explained in our responses to Rev 1 similar critique, coherence is a quantification of the similarity in frequency content of two signals as a function of frequency. As a consequence, we do not see any principled –theoretical or empirical– reason why coherence should not be applied in the context of click trains. However, we duly note both reviewers concerns and thank them for raising this issue, which has lead us refining and improving our analysis approach.

In the absence of a clear theoretical intuition in disfavour of our use of coherence measurement and aiming to provide a physiologically sound assessment of this issue, we have controlled how these inputs are processed at the cochlear level. To do so, we have rerun all coherence analyses using a model of the cochlear output of this waveform instead of the original sound waveform. Using a well-established model of cochlear filterbanks (Moore and Glasberg, 1981), we have transformed click trains stimuli into physiologically plausible waveforms, and tested again the coherence with neural data using the exact same approach as before. This approach is described in details in the Methods section as follows:

Online Methods (P4: lines 132-143): *"In order to best model the relationship between the stimulus and the brain response while avoid spurious effects potentially due to the coherence measure we used, we primarily transformed the waveform to model the peripheral transformation of the sound occurring at the cochlear level. Sound waveforms were transformed into a time-frequency representation (spectrogram)*

*using a filter-bank approach. Waveforms were filtered using 128 different linear-phase finite impulse response filters (512th order). Filters were designed to estimate critical bands (Moore and Glasberg, 1981) with centre frequencies logarithmically spanning the frequency space and corresponding equivalent rectangular bandwidths (according to the equation: $BW = 24.7 * (F * 4.37 + 1)$, where BW denotes bandwidth and F denotes a centre frequency in KHz (Moore, 1995). Each filter's output was then Hilbert transformed in order to extract the analytic amplitude and log-transformed. We then averaged the output of this filter-bank processing to provide a novel waveform that reflects the output of the cochlea (Moore and Glasberg, 1981). "*

As finally noted in this section (P4, Line 144):

"Note that although this approach was meant to realistically model the processes occurring at the cochlear level, not applying this transformation did not significantly change the results."

For example, why should there be the same low coherence at ≤ 20 Hz as there is above ≥ 130 Hz? The authors suggest that clicks start to fuse into a continuous percept around 130 Hz, but click frequencies ~ 20 Hz are very clearly perceptible, quite rapid... and as I have just synthesized such a click train myself, I can attest they are still rather annoying. The authors point out that the network activated by 40 Hz click trains are much more widespread than with 20 Hz: if this is true, I would suspect 20 Hz to CAC of a magnitude similar to 40 Hz, but perhaps more restricted to primary auditory areas. It would be helpful to show if this were the case.

We thank the reviewer for these interesting remarks.

First of all, it may have escaped the reviewer's attention that we originally provided an informative supplementary video, visually summarizing the spread and strength of the CAC effect at each frequency of stimulation. We dutifully encourage the reviewer watching this illuminating depiction of our experimental observations which notably confirm the reviewer' intuition that 20 Hz CAC is indeed spatially more restricted to early auditory areas. In addition to this, please note that Fig. 2b and c report the quantitative measurement of CAC effect at each frequency. We also provide an assessment of these responses in each sub-region of the temporal lobe in Fig 3c, which provide a more exhaustive depiction of CAC and HG effects.

The comparison between 20 Hz and 130 Hz CAC levels is well taken. It is difficult for us to provide a clear-cut interpretation of this interesting physiological observation. On the low-frequency side, we believe that CAC estimation might be affected by early, event related evoked components developing between two successive clicks. At 20 Hz the inter-click interval is 50 ms which allows early N20/P30 auditory components (Liegeois-Chauvel, 2004) to develop, which would not necessarily be possible above this stimulus frequency, i.e. when click are presented less than 25 ms apart. On the higher-frequency end (>130 Hz), a similar straightforward explanation would be that at this frequency, transients occur too frequently for brain responses to follow the stimulus with a one-to-one temporal mapping.

Finally, as noted by the reviewer, it is indeed likely that 20 Hz click trains may sound as aversive as 40 Hz ones to some individuals. There is indeed some interindividual variability of subjective ratings across participants (See Fig 1b, upper panel). This is actually the reason why we decided to provide a replication of these study in an independent set of participants (Fig. 1b lower panels). This study clearly replicates our findings, thereby supporting the relevance of using averaged ratings as an index of salience.

Perhaps more confidence in this analysis could be supported by testing coherence of the click trains with synthesized ECoG-like signals, or by shuffling the relationship between the real ECoG data and the corresponding click trains. Maybe the technique of spike-field coherence could be adapted to this situation as an alternative?

We thank the reviewer for these interesting remarks and suggestions.

Please note that all measurements (whether power or coherence) were performed not only during the peristimulus period, but were systematically compared to the same measurement performed during pre-stimulus baseline period of the same duration (Online Methods P4: lines 132-143). We believe that this approach is equivalent to the one proposed by the reviewer.

Regarding the spike-field coherence idea, it is an interesting and arguably valid intuition. If we understand the reviewer correctly, the idea would be to test whether the temporal arrangement of spikes (clicks) is reflected in the phase of brain responses at the frequency of the stimulation. However, given that the frequency of the stimulus changes considerably, we would obtain considerably more occurrences of “spikes” in the high frequency conditions, which might bias the comparisons of estimations of phase concentration between frequencies.

At any rate, and with all due respect, we may be missing something here but it is not fully clear to us why confidence in this analysis is at stake here, in particular given the strength and reliability of our observations across patients.

We sincerely hope that altogether, the new analyses and experimental controls provided, as well as the fact that we have doubled the number of patients included will convince the reader –as well as the reviewer– of the reliability of these results.

The authors should clarify whether the intracranial EEG electrode data were collapsed across subjects prior to statistical comparison, or if statistics were performed across subjects. Otherwise, the statistical procedures for the behavioral experiments appear sound, and correction for multiple comparisons with FDR is appropriate for the performed analyses.

Fair point. We apologise that this aspect was not clear in the previous version of the manuscript. This has been clarified in the Online Methods section (P4: lines 166-178)

Minor issues:

The definition of "efficiency" was not straightforward to understand; it would be appreciated to include the equation directly in the methods section of this manuscript. Instead, the manuscript refers to an earlier paper by the first author, but even there, its definition is only found in the supplementary material.

Fair enough. Please note that the experiments involving this measurement have been removed from the manuscript (see above).

The source code for the analyses has not been made available, as is required by Nature's policy. This would be especially helpful not only reproducing results but also to examine details of how important measures in the manuscript (like CAC) were computed.

Fair enough. We have created a Github where the relevant code and functions can be found. Please see main text P9, Line 306: "*Custom made software is available at <https://github.com/LucArnal/SoundOfSalience>.*" .

Figure 1a/1b: it seems that the y-axis should simply be labelled z-score? Only the circles represent efficiency, but RT and accuracy appear on the same plot.

The reviewer is correct. Please note that these figures have been removed from the manuscript (see above).

abstract: "but instead" -> "but also" ?

Corrected.

typo on page 7: "intracacial"

Corrected.

Reviewer #3 (Remarks to the Author):

Arnal et al. report a series of behavioral experiments and human intracranial EEG recordings that address the perceptual benefits of temporally salient auditory stimuli and the neural mechanisms underlying such benefits. They find that subjective unpleasantness, coherence between the acoustic signal and brain response (CAC), and coherence between different brain areas peak for 40Hz click train stimuli. Based on this result, Arnal et al. suggest that exogenous entrainment of intrinsic gamma band oscillations across sensory and attentional networks induces perceptual salience as well as subjective unpleasantness. This is interpreted in support of the Communication-through-coherence (CTC) hypothesis, which states that communication across brain areas relies on neuronal synchronization in the gamma frequency band.

The paper reports four behavioral experiments, which measure the effect of temporal

modulation rate (5 - 400 Hz) in tones and click trains onto stimulus detection in noise (Experiment 1), left/right stimulus localization (Experiment 2 for tones and Exp. 3 for clicks), and localization at different loudness levels (Experiment 4). Two additional experiments (5 and 6) tested the threshold at which click trains are perceived as a continuous stimulus (temporal sampling limit experiment) and showed that this threshold is intimately related to ratings of subjective unpleasantness of click trains (subj. unpleasantness experiment). Finally, an iEEG experiment explored the effects of temporal modulation rate (10 - 200 Hz) on neural responses to click trains.

I have several major concerns regarding the connection between the 7 experiments reported in this manuscript that undermine the interpretability of the results. Moreover, some statistical procedures need to be described in more details, and others appear inappropriate and might have lead to inflated significance (Exp 1- 4, listed further below).

We thank the reviewer for her/his careful reading and assessment of our work. We have duly noted the reviewer's concern regarding the apparent disconnection between the two parts of the manuscript experiments and have thoroughly reworked the manuscript in this regard (please see Introductory remarks at the outset of this rebuttal).

Please note also that in light of the three reviewers comments, for the sake of clarity of the revised manuscript and for the reader to better appreciate the links between behavioural and neural iEEG data, we have decided not to report anymore the first three behavioural experiments in the new version of the ms. We have made a series of substantial changes and additional analyses to improve our new manuscript, which now focuses on the behavioural and neural characterization of the aversiveness of temporally salience sounds. We are very grateful for these comments anyway and for the efforts made by the reviewer to assess this part of our work. Of course, we will take all these constructive critiques into account, should these results be described in another publication.

The first concern is about the relationship between temporal salience, subjective pleasantness, and roughness/continuity perception. Temporal salience is operationalized in Experiments 1- 4 through participants' perceptual performance (detection/localization) for stimuli with different modulation rates. These experiments show that performance is best within some, experiment-specific, modulation range. This range depends on the experiment, but in all four studies the optimal modulation frequency lies in the range of 100- 200 Hz. On the contrary, Experiments 5 and 6 show that unpleasantness peaks at 40 Hz in the roughness/discontinuity range. It remains unclear how unpleasantness relates to the salience measures introduced in Experiments 1-4. If both measure the same thing, why are best frequencies so different between Exp1-4 and Exp5-6? If they measure different processes - as suggested in the discussion section - how is unpleasantness related to temporal salience and what is the relevance of the iEEG experiment to the concepts introduced in Exp 1-4? Overall the relation between experiments 1-4 and experiments 5-7 remains unclear.

Agreed. Please note that the first set of experiments is not reported anymore in the new version of the manuscript (see point above).

The second major concern is the importance of the 40Hz frequency band for temporal salience. Unpleasantness, CAC, and CCC clearly peak for 40 Hz click trains. However, it remains unclear how this relates to temporal salience and processing efficiency, which were characterized in Experiments 1-4 and peaked at much higher frequencies, above 100 Hz and outside the gamma range. An alternative hypothesis would be that signal processing in noise or at low intensities, as tested in Exp. 1-4, actually corresponds to evoked responses to stimulus onset, which monotonically increased with modulation frequency (Fig. 3a+b). Unfortunately the iEEG experiment did not include modulation frequencies >400 Hz to test this hypothesis. It remains unclear how sound detection and localization relate to 40 Hz oscillations or the gamma band in general.

Thank you for pointing out these apparent discrepancies. Although we do believe that the notion of “temporal salience” had some merit to account for the observed results in the context of vocal communication efficiency, we agree that these differences make it difficult to describe all these results in a coherent framework. We have removed the first set of experiments from the new version of the manuscript (see point above) in order to provide a better assessment of the experiment for which we have both behavioural and neural data.

Overall this paper seems to contain two separate data sets: 1) Results on the temporal sampling limit in different perceptual tasks (Experiment 1- 4). 2) Neural signatures of subjective unpleasantness at different temporal modulation rates (Experiment 5-7).

Agreed. Please see above.

Detailed comments

Introduction

- P.2 2nd paragraph of introduction: This paper does not address the neural basis of the transition from roughness to pitch percept, as misleadingly suggested by this paragraph.

We agree with the reviewer’s remark that the previous version of the manuscript did not specifically address this question. We have reworked this paragraph so that it better introduces the framework and aims of our work, namely to provide a characterisation of subjective percepts of temporal salience (whether within or above the roughness range) and to determine the putative neural determinants of these effects on auditory aversion in the roughness and pitch ranges respectively.

- P.2: What is a strobe phenomenon? Here, and similarly in the discussion, the authors refer to findings from vision research that might not be known to the target audience of this paper. Please explain analogies to visual processing.

Agreed. We have clarified this as follows (P3, Lines 37-42):

“Fast repetitive modulations produce “temporally salient” flickering percepts (e.g. strobe lights, vibrators, and alarm sounds (Arnal et al., 2015)), which efficiently capture attention, generally induce rough and unpleasant sensations, and elicit avoidance (Helmholtz, 1863). Despite the high ecological relevance of such flickering stimuli, there is to our knowledge no existing operational definition of temporal salience and only limited experimental work accounting for the intriguing aversive sensation such auditory textures produce and the reactions they trigger.”

Results & Figures

Experiments 1 to 4

Statistical analysis of linear vs. polynomial effects (i.e. inverted u-shape) in behavioral analysis: The authors calculate the correlation between actual and predicted behavioral measures using a linear and a quadratic model. They then test for a difference between those correlations. Because the linear model is nested in the quadratic model, the correlation value for the quadratic model will always be at least as good as for the linear model.

The t-tests reported for the quadratic model fit in the results are thus trivial and do not indicate whether the quadratic fit reflects an improvement over the linear fit. A correct test of whether the quadratic term improves the model fit will take into account the degrees of freedom of each model, which are smaller for the quadratic model.

One way would be to report significance values for the regression coefficients for the linear and quadratic terms in the full (i.e. quadratic model). If the quadratic fit is better than the linear fit, the coefficient for the quadratic term will be significantly <0 (due to the inverted u-shape) across participants. The model could either be fit across participants with a linear mixed-effects approach, or using a hierarchical approach, whereby the within-participant beta-values are t-tested against 0 at the group level (similar to the group analysis approach taken in fMRI).

Without knowing the results of this analysis, I'm afraid that the results of the behavioral experiments 1-4 remain hard to interpret.

Thank you for these remarks. Please note that this set of experiments is not anymore reported in the new version of the manuscript, which now focuses on the subjective characterization of temporal salience using click trains. However, we will take into account and carefully address these relevant comments, should these experiments be described in another publication.

1) To identify the range of modulation frequencies that lead to best behavioral performance, it is necessary to identify the frequencies at which performance drops significantly below the peak performance level. Please add the corresponding pairwise comparisons to the results section.

Thank you for these remarks. Please see point above.

2) The exact best performance modulation frequencies are only explicitly stated in the discussion section, not in the results section. Please add them to the results section (backed by appropriate quantifications, see above).

Thank you for these remarks. Please see above.

Experiments 5-6:

Methods section does not provide details on fitting and statistical testing procedure for the sigmoidal curve of discreteness/continuity perception.

Thank you for noticing this unfortunate omission. Explanations have been added in the Online Methods section (P2 lines 56-51 and 62-73) for both sigmoidal (Fig. 1a) and linear (Fig. 1b) fitting procedures.

Again, please provide quantitative backup for the maximal unpleasantness range of 40-60 Hz.

We have corrected this (misleading) statement and added quantitative backup (and more detailed explanation, please see P2 lines 68-73) regarding the deviation from linearity of aversion responses in the roughness range.

Experiment 7 - iEEG recordings

I have a series of concerns regarding the statistical analyses of the iEEG data. Without these additional details it is hard to interpret the results. Page 7:

1) The temporal range used in analysis of evoked responses in HG is very broad (0-400), especially given the excellent temporal resolution of ecog. How does narrowing it down or using only a window around the peak alter the results?

We used this approach to be maximally inclusive (based on visual inspection of the data), and avoid potentially missing significantly activated 'onset' electrodes, but the reviewer's point is well taken. We have run the same analysis while focusing on responses in the gamma range in a smaller time-window 50–150 ms. This does not significantly change the results: a very similar subset of electrodes across patients, evidencing that the pattern of activation is robust and consistent regardless of the size of window of interest (correlations: with frequency: $r^2 = 0.611$, $p = 0.008$; with salience: $r^2 = 0.075$, $p = 0.512$).

2) No power is reported for analysis of coherence in different frequency ranges. Please report whether power in the relevant frequency bands was increased and discuss them with respect to the other results.

Agreed. We now consistently report power effects (and related correlations with stimulus energy and salience) accordingly per sub-region. Please see figures 3-6 panels' b-e.

3) Coherence analysis: Please explicitly explain in the methods section why coherence values are obtained timepoint by timepoint, rather than a single value per patient, electrode, and condition across the time course of the analysis window. Overall the description of the coherence metric is hard to follow.

Thank you for this remark. Coherence was actually measured across the whole time-window and not point-by-point, as mistakenly indicated. We have clarified the description and assessment of the coherence metric (P4, Lines 146-162), also in accordance with rev 2 and 3 remarks (please see introductory paragraph).

4) For example electrode 1 as well as for the group analysis, the maximal coherence is indicated in the 40-130 Hz range. However in Figure 3, the peak in coherence seems to be at 60Hz, whereas coherence appears at almost 0 at 130 Hz. Please indicate significance against 0 at different coherence levels if the exact range of stimulus frequencies at which coherence is maximal is important.

Thank you for this comment. We now consistently report statistical significance of the CAC effect at the group level at each frequency in Figure 2c.

5) Please indicate the overall number of electrodes in this analysis.

Good point. This information was indeed missing and we now systematically report numbers of activated electrodes in the figures.

Page 9 (Cortico-cortical coherence):

1) The frequency range of 30-80 Hz is very broad (>1 octave), and it is questionable whether oscillatory entrainment happens throughout this range or whether it reflects short bursts of activity outside the narrow band of 40-50 Hz, in which evidence for oscillatory processes is ample (e.g. in work by P. Fries).

Agreed. Although this is indeed an important issue, we do not aim at drawing strong conclusions with regards to “oscillatory” entrainment. Given that we present click trains, it is difficult to dissociate whether these effects indeed reflect “oscillatory” entrainment or just repetitive neural responses to these transients.

On the other hand, we do find intriguing –and worth discussing– the finding that exogenous entrainment is maximal at frequencies that roughly correspond to the gamma-band and high-frequency activities in the brain.

Therefore, we now refrain referring to “oscillatory” entrainment in the results section and only discuss the finding that stimuli in a restricted range induce sustained responses in a widespread network of regions, thereby supporting the notion that coherent responses in specific ranges may reflect the enhancement of neuronal communication.

2) What is the evidence/argument that favors communication between brain areas as source of coherence over coincidence based on the same rhythmic input? An

analysis that could alleviate this concern partial out the exogenous effects of the stimulus before testing for cortico-cortical coherence.

Point well taken. We do agree that it is hardly feasible to address this important issue. Therefore, and because it did not provide much relevant information with regards to our claims, we have removed the analysis that measured the synchronization between electrodes in this new version of our manuscript.

P18: 'We then transformed these data into time-courses of t-scores (calculated at each time point across trials) per contact, condition and participant. As a result, data were distributed normally, which allowed us to use standard parametric tests (e.g., paired t-test, repeated-measures ANOVA) to assess the statistical significance of observed effects.'

It is not clear what the authors mean here. If they are describing a linear normalization such as z-scoring (which seems to be the procedure used here, despite the name 't-scores'), z-scoring does not change the shape of the distribution, e.g. from skewed to normal. Rather, z-scoring only changes the range of values to allow for comparability across baselines. If they are referring to some non-linear transformation that changes the shape of data distribution to resemble a normal distribution, this needs to be described in more detail.

Fair enough. We understand this remark and apologise for the confusing statement, which has been removed. Please note that in order to account for potential SNR differences and comparability across baselines –as correctly pointed out by the reviewer– the data were individually transformed in t-statistic values (output of the `ttest.m` function in matlab, relevant matlab code is available on github). Although comparable for large sample size, this transformation is actually slightly different from z-scoring (see e.g. <https://www.statisticshowto.datasciencecentral.com/probability-and-statistics/hypothesis-testing/t-score-vs-z-score/>). We preferred this option because this transformation is valid even for small sample size (<30 trials, see e.g. webpage below). This has been clarified in the Online Methods section.

Minor comments:

P3: "We hypothesize that providing fast, but still perceptible, acoustic temporal cues". Likely means fast but still perceptible as distinct inputs and not a continuous stream. The wording is ambiguous.

We apologize for the confusion. We have slightly changed this sentence and now provide a clearer explanation, as follows:

Main text P3 Lines 43-47: *"We hypothesise that providing fast, but still discretisable and perceptible, temporally salient acoustic cues should enhance neural processing and ensuing aversive sensation. Such a strategy, however, is arguably constrained by the capacity of the auditory system to discretize – i.e. to faithfully encode and ultimately perceive– these temporal cues."*

P. 3: What is the roughness range? Please define and use consistently throughout the paper.

Fair enough. As also suggested by rev 1, we have provided clarifications as follows:

P1 Line 24: *“Another important emerging feature is roughness, an acoustic texture that arises from fast repetitive acoustic transients. Although the delimitation of the roughness range –whether psychoacoustic or perceptual– may slightly vary depending on experimental settings (Besser, 1967; Fastl and Zwicker, 2007; Krumbholz et al., 2000), empirical observations consistently suggest that sensory systems and perception are exceedingly well tuned to recurring temporal features in the 30–150 Hz range (Arnal et al., 2015; Helmholtz, 1863; Langner, 1992; Terhardt, 1974) “*

P2 line 37: *“Fast repetitive modulations produce “temporally salient” flickering percepts (e.g. strobe lights, vibrators, and alarm sounds (Arnal et al., 2015)), which efficiently capture attention, generally induce rough and unpleasant sensations, and elicit avoidance (Helmholtz, 1863) ”.*

P.4: 'Because of neuronal (or peripheral) adaptation mechanisms, perceptual efficiency is presumably only optimal at the onset of sensory transients and later becomes suboptimal, especially at low SNRs, when the risk of missing the sound onset are higher'

please provide a reference to the literature to back up this claim.

We apologise that this sentence was confusing and did not adequately reflect our logic. This sentence has been removed as it pertained to the behavioural experiments that are no longer reported in the new version.

Figures:

Figure 1:

Please add axes for accuracy and reaction time in panels a and b.

This figure has been removed (please see above).

Please change x-axis in panels a and b to scale. The categorical scale distorts the scale of the frequency effects and might induce a wrong percept of the curvature of the effects.

Also, it would be helpful to indicate (e.g. with vertical lines) the range in which performance is at best and does not differ between stimulus frequencies.

Fair enough, thank you for this remark. This figure has been removed from the manuscript (please see above).

Figure 3:
Panel b: Caption indicated that electrodes are localized to auditory areas, however,

electrodes are also marked in medial temporal (not really auditory anymore) and prefrontal areas. please report % electrodes in each area that responded to quantify this claim. From caption it is not clear whether this plot shows electrodes pooled across all patients in this study or only portions of the data. Please specify.

Fair enough. This sentence was confusing and unnecessary as we now provide a detailed characterisation and quantification of these effects on the basis of anatomical atlases. We now consistently report the total number of electrodes in each considered sub-regions according to anatomical or functional atlas-based labelling.

Panels a-c: Modulation frequency range appears to be 5-250 Hz, but methods section indicated 10-200. Please check and correct.

Corrected. Thanks for pointing this out.

Panel c: if little black dots indicate outliers, they are hardly visible. Also, they are not mentioned in caption.

Black dots (now represented as grey circles) indicate averages per patient. This is now specified in the legends.

Figure 4: The use of the term CAC in figure 4 is confusing because it's cortico-cortical and not cortico-acoustic coherence. Is this average coherence across time points?

Please specify.

Agreed. This figure has been removed from the manuscript (please see above).

References

- Arnal, L.H., Flinker, A., Kleinschmidt, A., Giraud, A.L., and Poeppel, D. (2015). Human Screams Occupy a Privileged Niche in the Communication Soundscape. *Curr. Biol.* *25*, 2051–2056.
- Besser, G.M. (1967). Some Physiological Characteristics of Auditory Flutter Fusion in Man. *Nature* *214*, 17–19.
- Destrieux, C., Fischl, B., Dale, A., and Halgren, E. (2010). Automatic parcellation of human cortical gyri and sulci using standard anatomical nomenclature. *Neuroimage*.
- Fastl, H., and Zwicker, E. (2007). Pitch and pitch strength. *Psychoacoustics Facts Model.* 111–148.
- Fisch, L., Privman, E., Ramot, M., Harel, M., Nir, Y., Kipervasser, S., Andelman, F., Neufeld, M.Y., Kramer, U., Fried, I., et al. (2009). Neural “Ignition”: Enhanced Activation Linked to Perceptual Awareness in Human Ventral Stream Visual Cortex. *Neuron* *64*, 562–574.
- Hayes, D.J., and Northoff, G. (2012). Common brain activations for painful and non-painful aversive stimuli. *BMC Neurosci.* *13*, 60.
- Helmholtz, H. von (1863). *On the Sensations of Tone as a Physiological Basis for the Theory of Music*. New York Dover.
- Krumbholz, K., Patterson, R.D., and Pressnitzer, D. (2000). The lower limit of pitch as determined by rate discrimination. *J. Acoust. Soc. Am.* *108*, 1170–1180.
- Langner, G. (1992). Periodicity coding in the auditory system X; *60*, 115–142.
- Liegeois-Chauvel, C. (2004). Temporal Envelope Processing in the Human Left and Right Auditory Cortices. *Cereb. Cortex* *14*, 731–740.
- Moore, B.C.J. (1995). Frequency Analysis and Masking. In *Hearing*, p.
- Moore, B.C.J., and Glasberg, B.R. (1981). Auditory filter shapes derived in simultaneous and forward masking. *J. Acoust. Soc. Am.*
- Oliveira, G., Davidson, A., Holczer, R., Kaplan, S., and Paretzky, A. (2016). A Comparison of the Use of Glottal Fry in the Spontaneous Speech of Young and Middle-Aged American Women. In *Journal of Voice*, (Mosby), pp. 684–687.
- Ray, S., Crone, N.E., Niebur, E., Franaszczuk, P.J., and Hsiao, S.S. (2008). Neural Correlates of High-Gamma Oscillations (60-200 Hz) in Macaque Local Field Potentials and Their Potential Implications in Electrocorticography. *J. Neurosci.* *28*, 11526–11536.
- Terhardt, E. (1974). On the perception of periodic sound fluctuations (roughness). *Acta Acust. United with Acust.*

Reviewers' comments:

Reviewer #1 (Remarks to the Author):

I am satisfied the author's responses to my concerns. Further, I believe the paper is much improved by its shorter length and more focused narrative. I continue to believe that this work provides the most compelling neurobiological account of auditory roughness to date and thus look forward to its publication, which I now recommend without hesitation. On reading through the edits on the manuscript, I have several minor suggestions for the author's to entertain, all aimed at increasing clarity.

ABSTRACT:

"artifactual" Suggestion: "artificial"

"...emitting fast but perceptible amplitude modulations in the roughness (20-150 Hz) range"

Suggestion:

"...emitting fast but perceptible amplitude modulations in the "roughness range" (20-150 Hz) "

"... suggesting that roughness enhances auditory aversion through the spreading of neural synchronisation". This supposes causality where only correlation is implied. Suggestion:

"... consistent with the hypothesis that roughness enhances auditory aversion through the spreading of neural"

"These results demonstrate that emitting rough sounds boosts perceptual salience by imposing neural communication through coherence in the receiver's brain". Confusing because of the ambiguous meaning of "communication" here, inter-individual vs. inter-neuronal. Suggestion: "These results demonstrate that emitting rough sounds boosts perceptual salience by imposing large-scale neural coherence in the receiver's brain"

RESULTS

L150 - "Building upon the observation that sustained responses to rough sounds do not confine to the super temporal auditory cortical regions, we then sought..."

Suggestion:

"Building upon the observation that sustained responses to rough sounds ARE not confined to super temporal auditory cortical regions, we NEXT sought..."

L156 - "We first investigated whether rough features might additionally recruit regions classically involved in the processing of aversive stimuli, namely subcortical and limbic regions." Add references. It would also help to be more specific about subcortical regions here (where they are first introduced), given that, generally speaking, "subcortical regions" are general involved in just about everything. I see that the next sentence includes the specific structures.... Could also say "key nodes in limbic and subcortical networks"

L172 - "infra-limbic"? This term is used to mean something different in the mouse literature... Better to avoid.

Reviewer #2 (Remarks to the Author):

The authors have substantially overhauled the manuscript, tightening the focus on the intracranial EEG experiment on auditory salience, and behavioral experiments that directly relate to the investigated hypotheses. Together with the additional details requested by the other reviewers and

myself, the manuscript now reads much more clearly and I appreciate the effort undertaken.

However, I still have major theoretical concerns about the use of cerebro-acoustic coherence with click train stimuli. CAC appears better suited to continuous signals, such as those of naturalistic acoustic stimuli. Indeed, it is most typically used with the envelope of speech signals, as in the original paper that introduced the term (Peelle, Gross, Davis, 2013) and all the papers that have appeared in the literature using the CAC technique, as well as related coherence approaches (as in Hyojin Park et al, 2015).

In my view, the spectral properties of clicks and click trains aren't well-suited to the CAC measure. A click closely approximates a delta function, which has a Fourier transform of 1 across all frequencies. Although this would deviate slightly for real-world discrete samples with typical auditory bandwidth, each individual click in a click train nevertheless has identical spectral content, while the click train itself is highly non-stationary. The relevant parameter in this study is of course the click train frequency (repetition rate), and certainly there would be some additional components of the Fourier transform that relate to it. But whether the FFT implicit in the coherence calculation would isolate that component with fidelity is highly dependent on the FFT parameters (such as DFT length and selected time window). While it may be possible to optimize the analysis for the click train frequency component, the provided code relies on the defaults for the mscohere function (and therefore the defaults for the underlying FFT operations).

In fact, applying the code to one of my own unrelated datasets (a simple visual flash-evoked response with no flicker or "flash train") also yields a ~40 Hz peak in the CAC profile (see attached figure). The finding of coherence with the 'click train' is of course coincidental, but probably arose due to phase-locked 40 Hz power solely in the visual response relative to baseline. The conclusions presented in this manuscript may still be correct -- it is likely that the iEEG electrodes that show a response of some kind are indeed the same ones that show elevated coherence over baseline, so they are responding to the click trains in some way -- but I'm afraid a different approach must be taken to be fully convincing of the effect.

Code for the Moore & Glasberg cochlear transformation was not provided, but I suspect that this procedure does not fully rectify this issue either, given the use of default FFT parameters in the coherence analysis.

It seems more logical to instead employ a frequency tagging approach in analyzing the iEEG responses. If the neural responses are a function of the clicktrain frequency, then this should be observable in the simple power spectrum of the electrodes showing activity and/or the intertrial coherence of those electrodes. These analyses would be more appropriate as the iEEG data is continuous with a more informative FFT than the stimuli themselves.

The authors have addressed my other concerns. The new text and revised figures are clear and easy to follow. This is a compelling narrative, but the authors have not convinced me that the use of CAC is appropriate. I hope the explanation of my reservations above clarify the problem and provide a constructive suggestion for a solution.

Some minor comments:

In discussing the ecological relevance, the authors may consider baby cries, which have tremor/vibrato components in the range examined in their study (Kent & Murray, 1982).

"Stimuli and Procedures" in online Methods: "square waves of 1 second duration" is a bit confusing, since the clicks themselves could be considered brief square waves. Perhaps clearer to simply say that the click train duration was 1 second.

Line 104 of Online Methods: was it truly a notch filter for removing powerline interference, or was

it FieldTrip's "DFT filter"?

Reviewer #3 (Remarks to the Author):

The authors have carefully addressed my previous concerns. Moreover, the new structure of the manuscript greatly improved readability, main questions and claims are now clear and compelling. I have a few remaining questions that need to be addressed before publication:

- 1) Overall, introduction and discussion are greatly improved, much clearer and easier to follow. One remaining suggestion is that it would be appropriate to briefly discuss the limits to the use of aversion as a proxy for salience.
- 2) Behavioral experiments (Figure 1):
 - Indicate in figure legends that light grey is single participant data
 - There is quite some variability in individual thresholds of continuity. Does the threshold correlate with the individual peak in aversion in the roughness range?
- 3) iEEG (Figure 2):
 - In response to one of my previous remarks, the authors clarified that CAC was calculated across the complete time window per electrode, condition and frequency band. However, Fig. 2a bottom panel shows a plot of CAC as function of time within a trial – please clarify the procedure used to obtain this panel.
- 4) It would be a valuable addition to the manuscript to see neural responses on an example electrode alongside the stimulus so that the reader could develop an intuition for what neural activity on an electrode with high CAC looks like. This could be a supplementary figure.

Rebuttal R2: Ms. No. NCOMMS-18-08959-T

(L.H. Arnal, A. Kleinschmidt, L. Spinelli, A.-L. Giraud, P. Mégevand)

General comments:

We thank the three reviewers for their positive and thoughtful comments that have helped us reworking important details of the manuscript. At the outset of our rebuttal, we would like to provide some general comments that are relevant to the various concerns of the reviewers.

We have taken into account all suggested edits and corrected the manuscript accordingly.

As reviewer 2 still expressed concerns with regards to the CAC methods, we have carefully assessed these issues 1- by modifying some of the parameters of the CAC analyses according to the reviewers' correct criticism and 2- by providing an additional control analysis using the suggested frequency-tagging approach. The results are compelling and suggest that although the reviewer's point was well taken, the two approaches yield very similar results and do not change the conclusions of the manuscript. We now report this additional control analysis in the methods section and provide a quantitative assessment of the similarity between the two approaches in a Supplementary figure.

We arbitrated that given the similarity in the outcomes of the two approaches, providing this comparison was methodologically valuable and validates the CAC approach. At any rate, please note that we have revised and reformatted the manuscript and figures using Rev 2 suggestion to correct the parameters of the CAC analysis.

Please find our point-by-point reply to the reviewers below.

Reviewers' comments:

Reviewer #1 (Remarks to the Author):

I am satisfied the author's responses to my concerns. Further, I believe the paper is much improved by its shorter length and more focused narrative. I continue to believe that this work provides the most compelling neurobiological account of auditory roughness to date and thus look forward to its publication, which I now recommend without hesitation. On reading through the edits on the manuscript, I have several minor suggestions for the author's to entertain, all aimed at increasing clarity.

ABSTRACT:

"artifactual" Suggestion: "artificial"

Thank you. Please note that this sentence has been removed from the abstract to comply with editorial length restrictions.

"...emitting fast but perceptible amplitude modulations in the roughness (20-150 Hz) range"

Suggestion:

"...emitting fast but perceptible amplitude modulations in the "roughness range" (20-150 Hz)"

Corrected.

"... suggesting that roughness enhances auditory aversion through the spreading of neural synchronisation". This supposes causality where only correlation is implied. Suggestion:

“... consistent with the hypothesis that roughness enhances auditory aversion through the spreading of neural”

Corrected.

“These results demonstrate that emitting rough sounds boosts perceptual salience by imposing neural communication through coherence in the receiver’s brain”. Confusing because of the ambiguous meaning of “communication” here, inter-individual vs. inter-neuronal.

Suggestion:

“These results demonstrate that emitting rough sounds boosts perceptual salience by imposing large-scale neural coherence in the receiver’s brain”

Thank you for this suggestion. This sentence has been removed from the abstract to comply with editorial length restrictions.

RESULTS

L150 - “Building upon the observation that sustained responses to rough sounds do not confine to the super temporal auditory cortical regions, we then sought...”

Suggestion:

“Building upon the observation that sustained responses to rough sounds ARE not confined to super temporal auditory cortical regions, we NEXT sought...”

Corrected.

L156 - “We first investigated whether rough features might additionally recruit regions classically involved in the processing of aversive stimuli, namely subcortical and limbic regions.” Add references. It would also help to be more specific about subcortical regions here (where they are first introduced), given that , generally speaking, “subcortical regions” are general involved in just about everything. I see that the next sentence includes the specific structures.... Could also say “key nodes in limbic and subcortical networks”

Thank you for these suggestions. We have added relevant references and changed the text according to the reviewer’s suggestion.

L172 - “infra-limbic”? This term is used to mean something different in the mouse literature... Better to avoid.

Corrected.

Reviewer #2 (Remarks to the Author):

The authors have substantially overhauled the manuscript, tightening the focus on the intracranial EEG experiment on auditory salience, and behavioral experiments that directly relate to the investigated hypotheses. Together with the additional details requested by the other reviewers and myself, the manuscript now reads much more clearly and I appreciate the effort undertaken.

Thank you for your positive comments.

We understand the reviewer’s concern and carefully considered these points. As described in what follows, we have tested the alternative approach proposed by the reviewer. As the reader may have similar concerns, we decided to evaluate and report how using the proposed method would affect our observations. It turns out that the new results fully confirm

the initial ones (see below), which, in our view, validates the use of CAC to measure steady-state responses. Please note that as we plan to use a similar approach to analyze brain responses to natural vocalizations, this additional control allows us to firmly build upon the current study for future works.

However, I still have major theoretical concerns about the use of cerebro-acoustic coherence with click train stimuli. CAC appears better suited to continuous signals, such as those of naturalistic acoustic stimuli. Indeed, it is most typically used with the envelope of speech signals, as in the original paper that introduced the term (Peelle, Gross, Davis, 2013) and all the papers that have appeared in the literature using the CAC technique, as well as related coherence approaches (as in Hyojin Park et al, 2015).

In my view, the spectral properties of clicks and click trains aren't well-suited to the CAC measure. A click closely approximates a delta function, which has a Fourier transform of 1 across all frequencies. Although this would deviate slightly for real-world discrete samples with typical auditory bandwidth, each individual click in a click train nevertheless has identical spectral content, while the click train itself is highly non-stationary. The relevant parameter in this study is of course the click train frequency (repetition rate), and certainly there would be some additional components of the Fourier transform that relate to it. But whether the FFT implicit in the coherence calculation would isolate that component with fidelity is highly dependent on the FFT parameters (such as DFT length and selected time window). While it may be possible to optimize the analysis for the click train frequency component, the provided code relies on the defaults for the `mscohere` function (and therefore the defaults for the underlying FFT operations).

We agree with the reviewer that the coherence approach was originally developed by Peelle et al. (now cited) to test the entrainment of brain responses to continuous signals such as speech. We use click trains as a simplified version of rough alarm vocalization, to isolate the putative ecological advantage of using rough temporal features to capture the listener's attention. The reason we selected the shape of these stimuli as square waves rather than using sinusoidal AM to test ASSRs is that we were concerned that not only the rate but also the rising slope of each cycle would increase as a function of the repetition frequency, and might hence introduce unwanted confounds.

The reviewer's criticism regarding the use of defaults parameters for the calculation of coherence is correct. We have changed this in our analysis by applying a larger window size and overlap value that is more adequate to perform this coherence analysis at low stimulation frequencies. We apologize for this mistake and we have revised and reformatted the manuscript and figures. We now provide code corrected using Rev 2 appropriate suggestion (<https://github.com/LucArnal/SoundOfSaliency/blob/master/CACoh.m>).

As for the choice of the CAC metric to assess steady-state responses, we understand the reviewer's concern and decided to directly compare the outcome of the CAC versus *frequency-tagging* approach. The correlation between the results obtained using the two methods is so strong ($r=0.901$, $p=9e-24$, see Fig. S1) that we arbitrated that it convincingly validates our approach, thereby supporting the generalizability of the CAC measures to non-speech stimuli. However, please note that we are willing to revise our analysis pipeline and reformat the whole manuscript and figures using the proposed alternative approach, should the reviewer not be fully satisfied.

As to justify our approach and the comparison of the two methods, we have appended the following explanations and supplementary figure:

- In the main text (Lines 166–185):

“Of note, we initially intended to use the CAC approach¹³ to measure the temporal relationship between neural responses and click trains waveform while taking into account the shape of these stimuli. However, because of the peculiar spectral content of clicks and despite the use of a cochlear filter bank to model the output of the cochlea (see previous section), it remained possible that the CAC method might not be best suited to capture these effects and may introduce unwanted confounds in the analysis. Therefore, we aimed to reproduce our basic CAC finding (Fig. 2c, bottom panel) using an alternative, frequency-tagging approach that does not depend on the stimulus shape. Importantly, all other analysis steps (e.g. baseline correction) were the same as those described earlier for CAC. Here, instead of measuring stimulus-brain coherence (CAC), we measured the power of neural responses filtered at the stimulus frequency. As for the extraction of HG activity, amplitude time series were computed by band-pass filtering the iEEG signal, but this time, at the stimulus frequency $F \pm 0.5$ Hz. The envelope of each narrow band signal was obtained by taking the absolute value of the analytic signal resulting from a Hilbert transform. As this method appeared to provide slightly less robust SNRs, significance threshold for frequency-tagging ‘activation’ was set at $p < 0.05$ (instead of $p < 0.01$ for CAC) after FDR correction across all electrodes. We then obtained a similar graph of t-values across frequencies and participants (Fig. S1a) as the one we previously obtained for CAC (Fig. 2c, bottom panel). Measuring correlations between the outcomes of each CAC and Frequency-tagging approaches on averaged (Fig. S1b) and individual (Fig. S1c) data, we demonstrated that the two approaches yielded very similar results, thereby validating the use of CAC measurements to robustly measure sustained steady-state responses.”

Supplementary Figure 1: Frequency-tagging approaches approximate CAC measurements.
a. Estimation of the power of neural responses in the late peristimulus time window [0.8–1.8s] as a function of stimulus rate using a frequency tagging approach (instead of CAC as shown in Fig. 2C, bottom graph). Each coloured data point reflects neural response power (averaged

across 'entrained' electrodes and participants, see Methods section) at the stimulus frequency. Grey circles correspond to individual data.

b. In order to compare the results obtained using CAC and Frequency-tagging approaches, we measured the correlation between the outcomes of each measurement type across participants and stimulation frequencies. The strong association between the two measurements (Pearson's $r^2 = 0.811$, i.e. more than 80% shared variance) suggests that the two approaches consistently capture similar neural steady-state effects.

c. A similar correlative approach was conducted at the individual level across all electrodes and frequencies to compare the outcomes of CAC and frequency tagging approaches. This analysis reveals that the two methods provide highly similar results. Qualitatively, these observations further suggest that correlation strength between these two approaches depends on individual signal-to-noise ratios (i.e. is highest in participants with high t-values ranges).

In fact, applying the code to one of my own unrelated datasets (a simple visual flash-evoked response with no flicker or "flash train") also yields a ~40 Hz peak in the CAC profile (see attached figure). The finding of coherence with the 'click train' is of course coincidental, but probably arose due to phase-locked 40 Hz power solely in the visual response relative to baseline. The conclusions presented in this manuscript may still be correct -- it is likely that the iEEG electrodes that show a response of some kind are indeed the same ones that show elevated coherence over baseline, so they are responding to the click trains in some way -- but I'm afraid a different approach must be taken to be fully convincing of the effect.

We thank the reviewer for her/his careful assessment and for spending time and thoughtful efforts to characterize the putative origin of these effects. It is indeed quite common to observe a burst of evoked (phase-locked) gamma activity following the onset of a stimulus, which should in turn, result in a transient increase of CAC at this frequency. One would arguably expect that the frequency-tagging approach would yield a similar observation.

As stated in the manuscript (L110-114), we circumvented the potential confound of a transient onset phase-locking effect by applying "*the Cerebro-Acoustic Coherence (CAC, see Methods) between sounds and brain responses in the late time window (restricted to [0.8–1.8 s] to avoid potential contamination by onset and offset responses)*".

Code for the Moore & Glasberg cochlear transformation was not provided, but I suspect that this procedure does not fully rectify this issue either, given the use of default FFT parameters in the coherence analysis.

This is a fair point. We have corrected the code according to the reviewer correct criticisms. We also apologize that this part of the code was not available in our previous versions. We now provide the code for cochlear transformation together with the other functions –CAC and frequency-tagging– on Github.

It seems more logical to instead employ a frequency tagging approach in analyzing the iEEG responses. If the neural responses are a function of the clicktrain frequency, then this should be observable in the simple power spectrum of the electrodes showing activity and/or the intertrial coherence of those electrodes. These analyses would be more appropriate as the iEEG data is continuous with a more informative FFT than the stimuli themselves.

The reviewer is right, and as she/he correctly predicted, we do confirm that the frequency tagging approach reveals frequency selective effects, that are very similar to our initial observations (please see previous points).

The authors have addressed my other concerns. The new text and revised figures are clear and easy to follow. This is a compelling narrative, but the authors have not convinced me that

the use of CAC is appropriate. I hope the explanation of my reservations above clarify the problem and provide a constructive suggestion for a solution.

We would like to thank again the reviewer for her/his thoughtful and constructive efforts. We hope that the controls provided –as well as the striking similarity between the results using the two approaches– are both convincing and satisfying. If not, we apologize in advance to the three reviewers for the additional work and will consider modifying all the figures and text according to her/his requirements.

Some minor comments:

In discussing the ecological relevance, the authors may consider baby cries, which have tremor/vibrato components in the range examined in their study (Kent & Murray, 1982).

Reference added. Thank you for pointing this out.

"Stimuli and Procedures" in online Methods: "square waves of 1 second duration" is a bit confusing, since the clicks themselves could be considered brief square waves. Perhaps clearer to simply say that the click train duration was 1 second.

Corrected.

Line 104 of Online Methods: was it truly a notch filter for removing powerline interference, or was it FieldTrip's "DFT filter"?

Good point. This has been corrected in the Methods section.

Reviewer #3 (Remarks to the Author):

The authors have carefully addressed my previous concerns. Moreover, the new structure of the manuscript greatly improved readability, main questions and claims are now clear and compelling.

We thank the reviewer for these thoughtful comments on our manuscript.

I have a few remaining questions that need to be addressed before publication:

1) Overall, introduction and discussion are greatly improved, much clearer and easier to follow. One remaining suggestion is that it would be appropriate to briefly discuss the limits to the use of aversion as a proxy for salience.

Thanks for the suggestion. We have added the following sentence to the main text:

Line 44: *“Of note, although salience may not systematically result in aversive percept, we argue that in this specific context, temporal salience –owing to the imperative effect of exogenously saturating perceptual systems in time– constitutes a valid proxy of aversion.”*

2) Behavioral experiments (Figure 1):

- Indicate in figure legends that light grey is single participant data

Information added.

- There is quite some variability in individual thresholds of continuity. Does the threshold correlate with the individual peak in aversion in the roughness range?

Thank you for this interesting remark. Indeed we originally aimed at testing the hypothesis that individual transition values might be connected to individual aversive profiles. However, although there seemed to be a trend in the data, the results of this analysis were rather inconclusive and instable, depending too strongly on how we extracted relevant parameters (individual aversion peaks) for the correlation. Therefore, we decided not to report these qualitative observations, as they were insufficiently grounded statistically.

3) iEEG (Figure 2):

- In response to one of my previous remarks, the authors clarified that CAC was calculated across the complete time window per electrode, condition and frequency band. However, Fig. 2a bottom panel shows a plot of CAC as function of time within a trial – please clarify the procedure used to obtain this panel.

Good point, thank you for pointing this out. This information was indeed missing and has now been added in the Methods section (Lines 162–165) as follows:

“In order to qualitatively show how CAC evolves in time as a function of stimulus frequency for illustrative purposes (Fig. 2a, bottom panel), we also applied this method in a single electrode using a sliding 1000-ms window in 10-ms steps across the whole peristimulus time-course.”

4) It would be a valuable addition to the manuscript to see neural responses on an example electrode alongside the stimulus so that the reader could develop an intuition for what neural activity on an electrode with high CAC looks like. This could be a supplementary figure.

Thank you for this thoughtful suggestion. We have added an additional panel depicting the unfiltered ERP average of neural responses per condition in Fig. 2a (lower left panel). The panel also illustrates a click train (here 40 Hz) aligned with the onset of neural responses.

REVIEWERS' COMMENTS:

Reviewer #2 (Remarks to the Author):

The authors have presented a comparison of their CAC approach with frequency tagging that addresses my major concern. They have also satisfactorily addressed the minor issues that I pointed out last time.

I thought the earlier title "The sound of salience: how roughness enhances aversion through neural synchronisation" had a great ring to it and am not sure why the authors changed it for this revision. Both titles are of course fine, just wanted to provide that feedback. :-)

Thanks to the authors for working so hard on this to address my concerns, and for making their code available and comprehensible!

Reviewer #3 (Remarks to the Author):

The authors made a great effort addressing all of my concerns and with the additional clarifications provided in this revision I believe the manuscript is suitable for publication.

The frequency tagging analysis is an important addition to the more complex CAC analysis and I strongly support that it be mentioned in the manuscript and the figure be either in the SOM or a figure in the main text.